# Efficient Algorithms for Learning Depth-2 Neural Networks with General ReLU Activations

**Pranjal Awasthi**
Google Research
pranjalawasthi@google.com

**Alex Tang**
Northwestern University
alextang@u.northwestern.edu

**Aravindan Vijayaraghavan**
Northwestern University
aravindv@northwestern.edu

## Abstract

We present polynomial time and sample efficient algorithms for learning an unknown depth-2 feedforward neural network with *general* ReLU activations, under mild non-degeneracy assumptions. In particular, we consider learning an unknown network of the form $f(x) = a^\mathsf{T}\sigma(W^\mathsf{T}x + b)$, where $x$ is drawn from the Gaussian distribution, and $\sigma(t) \coloneqq \max(t, 0)$ is the ReLU activation. Prior works for learning networks with ReLU activations assume that the bias $b$ is zero. In order to deal with the presence of the bias terms, our proposed algorithm consists of robustly decomposing multiple higher order tensors arising from the Hermite expansion of the function $f(x)$. Using these ideas we also establish identifiability of the network parameters under minimal assumptions.

## 1 Introduction

The empirical success of deep learning in recent years has led to a flurry of recent works exploring various theoretical aspects of deep learning such as learning, optimization and generalization. A fundamental question in the theory of deep learning is to identify conditions under which one can design provably time-efficient and sample-efficient learning algorithms for neural networks. Perhaps surprisingly, even for the simplest case of a depth-2 feedforward neural network, the learning question remains unresolved. In this work we make progress on this front by studying the problem of learning an unknown neural network of the form

$$y = f(x) = a^\mathsf{T}\sigma(W^\mathsf{T}x + b). \tag{1}$$

We are given access to a finite amount of samples of the form $(x_i, y_i)$ drawn i.i.d. from the data distribution, where each $x_i$ is comes from the standard Gaussian distribution $\mathcal{N}(0, I)$, and $y_i = f(x_i)$. The goal is to design an algorithm that outputs an approximation of the function $f$ up to an arbitrary error measured in the expected mean squared metric (squared $\ell_2$ loss). An efficient learning algorithm has running time and sample complexity that are polynomial in the different problem parameters such as the input dimensionality, number of hidden units, and the desired error.

Without any further assumptions on the depth-2 network, efficient learning algorithms are unlikely. The recent work of [DKKZ20] provides evidence by proving exponential statistical query lower bounds (even when $x$ is Gaussian) that rule out a broad class of algorithms.

Several recent works have designed efficient algorithms for depth-2 neural networks in the special setting when the bias term $b = 0$. One prominent line of work [GLM18, BJW19] give polynomial time algorithms under the non-degeneracy assumption that the matrix $W$ has full-rank. Another body of work relaxes the full-rank assumption by designing algorithms that incur an exponential dependence on the number of hidden units [DKKZ20, CKM20, GK19], or a quasipolynomial dependence when the coefficients $\{a_i : i \in [m]\}$ are all non-negative [DK20]. There is little existing literature on learning neural networks in the presence of the bias term. A notable exception is an approach based on computing "score functions" [JSA15] that applies to certain activations with bias and requires

35th Conference on Neural Information Processing Systems (NeurIPS 2021).

various assumptions that are not satisfied by the ReLU function. The diminished expressivity of neural networks without the bias terms leads to the following compelling question:

*Can we design polynomial time algorithms even in the presence of bias terms in the ReLU units?*

We answer the question affirmatively by designing efficient algorithms for learning depth-2 neural networks with general ReLU activations, under the assumption that $W_{d \times m}$ has linearly independent columns (hence $m \leq d$). In fact, our algorithms can be extended to work under much weaker assumptions on $W$, that allow for $m \gg d$ ($m \leq O(d^{\ell})$ for any constant $\ell \geq 1$) in a natural smoothed analysis setting considered in prior works [GKLW19] (see Theorem 3.2 and Corollary 3.3). An important consequence of our techniques is the fact that the network parameters are identifiable up to signs, as long as no two columns of $W$ are parallel, and all the $\{a_i : i \in [m]\}$ are non-zero. Furthermore we show that this ambiguity in recovering the signs is inherent unless stronger assumptions are made.

**Conceptual and technical challenges with bias terms.** Similar to prior works [GKLW19, JSA15], our techniques rely on the use of tensor decomposition algorithms to recover the parameters of the network. In the absence of any bias, it can be shown that the 4th Hermite coefficient of the function $f(x)$ takes the form $\hat{f}_4 = \sum_{i=1} a_i w_i^{\otimes 4}$ where $w_i$ are the columns of $W$ and $a = (a_1, a_2, \dots, )$. When the columns of $W$ are linearly independent, existing algorithms for tensor decompositions in the full-rank setting can be used to recover the parameters [Har70].[1] However, when bias terms are present, there are several challenges that we highlight below.

In the presence of biases the $k$th Hermite expansion of $f(x)$ takes the form $\hat{f}_k = \sum_{i=1}^{m} a_i g_k(b_i) w_i^{\otimes k}$, where $g_k$ is a function that may vanish on some of the unknown $b_i$ parameters. This creates a hurdle in recovering the corresponding $w_i$. A simple example where the above approach fails is when some of the $b_i = \pm 1$, since the corresponding rank-1 terms vanish from the decomposition of $\hat{f}_4$. To overcome this obstacle, we first give a precise expression for the function $g_k(b_i)$ involving the $(k-2)$th Hermite polynomial (see Lemma 3.5). We then design an algorithm that decomposes multiple tensors obtained from Hermite coefficients to recover the parameters. We use various properties of Hermite polynomials to analyze the algorithm e.g., the separation of roots of consecutive Hermite polynomials is used to argue that each $w_i$ is recovered from decomposing at least one of the tensors.

Secondly, in the presence of the bias terms, recovery of all the parameters (even up to sign ambiguities) may not even be possible from polynomially many samples. For instance, consider a particular hidden node with output $\sigma(w_i^{\top} x + b_i)$. If $b_i$ is a large positive number then it behaves like a linear function (always active). Hence if multiple $b_i$s are large positive numbers then one can only hope to recover a linear combination of their corresponding weights and biases. On the other hand if $b_i$ is a large negative constant then the activation is 0 except with probability exponentially small in $|b_i|$. We cannot afford a sample complexity that is exponential in the magnitude of the parameters. Furthermore, when the columns of $W$ are not linearly independent, the tensor decomposition based method will only recover good approximations up to a sign ambiguity for the terms whose bias does not have very large magnitude i.e., we recover $\pm(w_i^{\top} x + b_i)$ if $|b_i|$ is not very large.

To handle the above issue we proceed in two stages. In the first stage we recover the network parameters (up to signs) of all the "good" terms, i.e., hidden units with biases of small magnitude. To handle the "bad" terms (large magnitude bias) we show that a linear functions is a good approximation to the residual function comprising of the bad terms. Based on the above, we show that one can solve a truncated linear regression problem to learn a function $g(x)$ that achieves low mean squared error with respect to the target $f(x)$. The output function $g(x)$ is also a depth-2 ReLU network with at most two additional hidden units than the target network.

There are several other technical challenges that arise in the analysis sketched above, when there are sampling errors due to using only a polynomial amount of data (for example, the tensors obtained from $\hat{f}_k$ may have some rank-1 terms that are small but not negligible, that may affect the robust recovery guarantees for tensor decompositions). We obtain our robust guarantees by leveraging many useful properties of Hermite polynomials, and a careful analysis of how the errors propagate.

---

[1] Tensor decompositions will in fact recover each ReLU activation up to an ambiguity in the sign. However, in the full-rank setting, the correct sign can also be recovered (as we demonstrate later in Theorem 3.1).

The rest of the paper is organized as follows. We present preliminaries in Section 1.1 followed by related work in Section 2. We then formally present and discuss our main results in Section 3. In Section 4 we present our main algorithm and analysis in the population setting, i.e., under the assumption that one has access to infinite data from the distribution. We then present the finite sample extension of our algorithm in Section 5 that achieves polynomial runtime and sample complexity.

## 1.1 Model Setup and Preliminaries

We consider the supervised learning problem with input $x \in \mathbb{R}^d$ drawn from a standard $d$-dimensional Gaussian distribution $\mathcal{N}(0, I_{d \times d})$ and labels $y$ generated by a neural network $y = f(x) = a^\mathsf{T} \sigma(W^\mathsf{T} x + b)$, where $a, b \in \mathbb{R}^m$, $W \in \mathbb{R}^{d \times m}$ and $\sigma$ is the element-wise ReLU activation function, i.e., $\sigma(t) = \max(t, 0)$. We denote the column vectors of $W$ as $w_i \in \mathbb{R}^d$ with $i \in [m]$ and $a_i$ as the $i$'th element of vector $a$, similarly for $b$ and $x$. We pose a constraint on magnitudes of $a, b, W$ such that they are all $B$-bounded for some $1 \leq B \leq \text{poly}(m, d)$, i.e. $\|a\|_\infty, \|b\|_\infty, \|W\|_\infty \leq B$, and $\min_{i \in [m]} |a_i| \geq 1/B$. Furthermore, we assume $\|w_i\|_2 = 1$ without loss of generality. If $w_i$ are not unit vectors, we can always scale $a_i$ and $b_i$ to $\|w_i\| a_i$ and $\frac{b_i}{\|w_i\|}$ respectively so that $w_i$ are normalized. We will denote by $\Phi(\cdot)$ the cumulative density function (CDF) of the standard Gaussian distribution. Finally, for a matrix $M$, we will use $s_k(M)$ to denote the $k$th largest singular value of $M$.

For introduction and further preliminaries regarding $\text{poly}(\cdot)$ notation, basics of Hermite polynomials and tensor decomposition, please refer to Appendix A.

## 2 Related Work

By now there is a vast literature exploring various aspects of deep learning from a theoretical perspective. Here we discuss the works most relevant in the context of our results. As discussed earlier, the recent works of [GLM18, BJW19, GKLW19] provide polynomial time algorithms for learning depth-2 feedforward ReLU networks under the assumption that the input distribution is Gaussian and that the matrix $W$ is full rank. Some of these works consider a setting where the output is also a high dimensional vector [BJW19, GKLW19], and also consider learning beyond the Gaussian distribution. However, these works do not extend to the case of non-zero bias.

The work of [JSA15] proposed a general approach based on tensor decompositions for learning an unknown depth-2 neural network that could also handle the presence of the bias terms. The tensor used in the work of [JSA15] is formed by taking the weighted average of a "score" function evaluated on each data point. In this way their approach generalizes to a large class of distributions provided one has access to the score function. However, for most data distributions computing the score function is a hard task itself. When the input distribution is Gaussian, then the score functions correspond to the Hermite coefficients of the target function $f(x)$ and can be evaluated efficiently. However, the analysis in [JSA15] does not extend to the case of ReLU activations for several reasons. Their technique needs certain smoothness and symmetry assumptions on the activations that do not hold for ReLU. These assumptions also ensure that all the terms in the appropriate tensor are non-zero. We do not make such assumptions, and tackle one of the main challenges by showing that one can indeed recover a good approximation to the network by analyzing multiple higher order tensors. Furthermore, the authors in [JSA15] assume that the biases, and the spectral norm of $W$ are both bounded by a constant. As a result they do not handle the case of biases of large magnitude where some of the ReLU units mostly function as linear functions (with high probability).

There have also been works on designing learning algorithms for neural networks without assumptions on the linear independence of columns of $W$. These results incur an exponential dependence on either the input dimensionality or the number of parameters in the unknown network [DKKZ20, CKM20], or quasipolynomial dependence when the coefficients $\{a_i : i \in [m]\}$ are all non-negative [DK20]. In particular, the result of [CKM20] provides a learning algorithm for arbitrary depth neural networks under the Gaussian distribution with an "FPT" guarantee; its running time is polynomial in the dimension, but exponential in the number of ReLU units. Given the recent correlational statistical query lower bounds on learning deep neural networks [DKKZ20, GGJ+20, DV21], getting a fully polynomial time algorithm without any assumptions is a challenging open problem, even under Gaussian marginals.

Polynomial time algorithms with fewer assumptions and beyond depth-2 can be designed if the activation functions in the first hidden layer are sigmoid functions [GK19]. Finally, there is also extensive literature on analyzing the convergence properties of gradient descent and stochastic gradient descent for neural networks. The results in this setting implicitly or explicitly assume that the target function is well approximated in the Neural Tangent Kernel (NTK) space of an unknown network. Under this assumption these results show that gradient descent on massively overparameterized neural networks can learn the target [APVZ14, Dan17, DFS16, AZLS19, DZPS19, ADH+19, LXS+19, CB18, JGH21].

## 3  Main Results

There are two related but different goals that we consider in learning the ReLU network:

- *Achieves low error:* Output a ReLU network $g(x) = a'^\top \sigma(W'^\top x + b')$ such that the $L_2$ error is at most $\varepsilon$ for a given $\varepsilon > 0$ i.e., $\mathbb{E}_{x \sim \mathcal{N}(0, I_{d \times d})}[(f(x) - g(x))^2] \leq \varepsilon^2$.

- *Parameter recovery:* Output $\widetilde{W}, \widetilde{a}, \widetilde{b}$, such that each parameter is $\varepsilon$-close (up to permuting the $m$ co-ordinates of $a, b \in \mathbb{R}^m$ and reordering the corresponding columns of $W$).

We remark that the second goal is harder and implies the first; in particular, when $\varepsilon = 0$, the second goal corresponds to identifiability of the model. However in some cases, parameter recovery may be impossible to achieve (see later for some examples) even though we can achieve the goal of achieving low error. As we have seen earlier, given $N$ samples if $b_i \gg \sqrt{\log N}$, then $\sigma(w_i^\top x + b_i)$ will be indistinguishable from the linear function $w_i^\top x + b$ w.h.p.; hence if there are multiple such $b_i \in [m]$ with large magnitude, the best we can hope to do is recover the sum of all those linear terms. Our first result shows that this is the only obstacle when we are in the *full-rank* or *undercomplete* setting i.e., $\{w_i : i \in [m]\}$ are linearly independent (in a robust sense).

**Theorem 3.1** (Full-rank setting). *Suppose $\varepsilon \in (0, 1)$ and $N \geq \text{poly}(m, d, 1/\varepsilon, 1/s_m(W), B)$ samples be generated by a ReLU network $f(x) = a^\top \sigma(W^\top x + b)$ that is $B$-bounded, and $|b_i| < c\sqrt{\log(1/\varepsilon mdB)}$ for all $i \in [m]$. Then there exists an algorithm that runs in $\text{poly}(N, m, d)$ time and with high probability recovers $\widetilde{a}_i, \widetilde{b}_i, \widetilde{w}_i$ such that $\|w_i - \widetilde{w}_i\|_2 + |a_i - \widetilde{a}_i| + |b_i - \widetilde{b}_i| < \varepsilon$ for all $i \in [m]$.*

The above theorem recovers all the parameters when the biases $\{b_i : i \in [m]\}$ of each ReLU unit does not have very large magnitude. Moreover even when there are $b_i$ of large magnitude, we can learn a depth-2 ReLU network $g$ that achieves low error, and simultaneously recover parameters for the terms that have a small magnitude of $b_i$ (up to a potential ambiguity in signs). In fact, our algorithm and guarantees are more general, and can operate under the much milder condition that $\{w_i^{\otimes \ell} : i \in [m]\}$ are linearly independent for any constant $\ell \geq 1$; the setting when $\ell > 1$ corresponds to what is often called the *overcomplete setting*. In what follows, for any constant $\ell \in \mathbb{N}$ we use $\text{poly}_\ell(n_1, n_2, \dots)$ to denote a polynomial dependency on $n_1, n_2, \dots$, and potentially exponential dependence on $\ell$.

**Theorem 3.2.** *Suppose $\ell \in \mathbb{N}$ be a constant, and $\varepsilon > 0$. If we are given $N$ i.i.d. samples as described above from a ReLU network $f(x) = a^\top \sigma(W^\top x + b)$ that is $B$-bounded then there is an algorithm that given $N \geq \text{poly}_\ell(m, d, 1/\varepsilon, 1/s_m(W^{\odot \ell}), B)$ runs in $\text{poly}(N, m, d)$ time and with high probability finds a ReLU network $g(x) = a'^\top \sigma(W'^\top x + b')$ with at most $m + 2$ hidden units, such that the $L_2$ error $\mathbb{E}_{x \sim \mathcal{N}(0, I_{d \times d})}[(f(x) - g(x))^2] \leq \varepsilon^2$. Furthermore there are constants $c = c(\ell) > 0, c' > 0$ and signs $\xi_i \in \{\pm 1\} \; \forall i \in [m]$, such that in $\text{poly}(N, m, d)$ time, for all $i \in [m]$ with $|b_i| < c\sqrt{\log(1/(\varepsilon \cdot mdB))}$, we can recover $(\widetilde{a}_i, \widetilde{w}_i, \widetilde{b}_i)$, such that $|a_i - \widetilde{a}_i| + \|w_i - \xi_i \widetilde{w}_i\|_2 + |b_i - \xi_i \widetilde{b}_i| < c' \varepsilon / (mB)$.*

In the special case of $\ell = 1$ in Theorem 3.2, we need the least singular value $s_m(W) > 0$ (this necessitates that $m \leq d$). This corresponds to the *full-rank setting* considered in Theorem 3.1. In contrast to the full-rank setting, for $\ell > 1$ we only require that the set of vectors $w_1^{\otimes \ell}, w_2^{\otimes \ell}, \dots, w_m^{\otimes \ell}$ are linearly independent (in a robust sense), which one can expect for much larger values of $m$ typically. The following corollary formalizes this in the smoothed analysis framework of Spielman and Teng [ST04], which is a popular paradigm for reasoning about non-worst-case instances [Rou20]. Combining the above theorem with existing results on smoothed analysis [BCPV19] implies polynomial time

learning guarantees for non-degenerate instances with $m = O(d^\ell)$ for any constant $\ell > 0$. Below, $\widehat{W}$ denotes the columns of $W$ are $\tau$-smoothed i.e., randomly perturbed with standard Gaussian of average length $\tau$ that is at least inverse polynomial (See Section E for the formal smoothed analysis model and result).

**Corollary 3.3** (Smoothed Analysis). *Suppose $\ell \in \mathbb{N}$ and $\varepsilon > 0$ are constants in the smoothed analysis model with smoothing parameter $\tau > 0$, and also assume the ReLU network $f(x) = a^\top \sigma(\widehat{W}^\top x + b)$ is $B$-bounded with $m \leq 0.99\binom{d+\ell-1}{\ell}$. Then there is an algorithm that given $N \geq \text{poly}_\ell(m, d, 1/\varepsilon, B, 1/\tau)$ samples runs in $\text{poly}(N, m, d)$ time and with high probability finds a ReLU network $g(x) = a'^\top \sigma(W'^\top x + b')$ with at most $m + 2$ hidden units, such that the $L_2$ error $\mathbb{E}_{x \sim \mathcal{N}(0, I_{d \times d})}[(f(x) - g(x))^2] \leq \varepsilon^2$. Furthermore there are constants $c, c' > 0$ and signs $\xi_i \in \{\pm 1\} \; \forall i \in [m]$, such that in $\text{poly}(N, m, d)$ time, for all $i \in [m]$ with $|b_i| < c\sqrt{\log(1/(\varepsilon \cdot mdB))}$, we can recover $(\widetilde{a}_i, \widetilde{w}_i, \widetilde{b}_i)$, such that $|a_i - \widetilde{a}_i| + \|w_i - \xi_i \widetilde{w}_i\|_2 + |b_i - \xi_i \widetilde{b}_i| < c'\varepsilon/(mB)$.*

While our algorithm and the analysis give guarantees that are robust to sampling errors and inverse polynomial error, even the non-robust analysis has implications and, implies identifiability of the model (up to ambiguity in the signs) as long as no two rows of $W$ are parallel. Note that in general identifiability may not imply any finite sample complexity bounds.

**Theorem 3.4** (Partial Identifiability). *Suppose we are given samples from a ReLU network $f(x) = a^\top \sigma(W^\top x + b)$ where $\min_{i \in [m]} |a_i| > 0$ and no two columns of $W$ are parallel to each other. Then given samples, the model parameters $(a_i, b_i, w_i : i \in [m])$ are identified up to ambiguity in the signs and reordering indices i.e., we can recover $\{(a_i, \xi_i b_i, \xi_i w_i) : i \in [m]\}$ for some $\xi_i \in \{+1, -1\} \; \forall i \in [m]$.*
*Moreover given any $(\xi_i \in \{+1, -1\} : i \in [m])$ such that*

$$\sum_{i=1}^m a_i \Phi(\xi_i b_i) \xi_i w_i = \sum_{i=1}^m a_i \Phi(b_i) w_i, \quad \text{and} \quad \sum_{i=1}^m a_i b_i \Phi(\xi_i b_i) = \sum_{i=1}^m a_i b_i \Phi(b_i), \qquad (2)$$

*we have that the set of parameters $((a_i, \xi_i b_i, \xi_i w_i : i \in [m])$ also gives rise to the same distribution.*

The above theorem shows that under a very mild assumption on $W$, the parameters can be identified up to signs. However, this ambiguity in the signs may be unavoidable – the second part of the Theorem 3.4 shows that any combination of signs that match the zeroth and first Hermite coefficient gives rise to a valid solution (this corresponds to the $d + 1$ equations in (2)). Even in the case when all the $b_i = 0$, we have non-identifiability due to ambiguities in signs whenever the $\{w_i : i \in [m]\}$ are not linearly independent for an appropriate setting of the $\{a_i\}$; see Claim 4.8 for a formal statement. On the other hand, Theorem 4.1 gives unique identifiability result in the full-rank setting (as there is only one setting of the signs that match the first Hermite coefficient in the full-rank setting).

Our results rely on the precise expressions for higher order Hermite coefficients of $f(x)$ given below.

**Lemma 3.5.** *Let $\hat{f}_k = \mathbb{E}_{x \sim \mathcal{N}(0, I)}[f(x) He_k(x)]$ (with $k \in \mathbb{N}$) be the $k$'th Hermite coefficient (this is an order-$k$ tensor) of $f(x) = a^\top \sigma(W^\top x + b)$. Then*

$$\hat{f}_0 = \sum_{i=1}^m a_i \left( b_i \Phi(b_i) + \frac{\exp(-\frac{b_i^2}{2})}{\sqrt{2\pi}} \right), \quad \hat{f}_1 = \sum_{i=1}^m a_i \Phi(b_i) w_i \qquad (3)$$

$$\forall k \geq 2, \quad \hat{f}_k = \sum_{i=1}^m (-1)^k \cdot a_i \cdot He_{k-2}(b_i) \cdot \frac{\exp(\frac{-b_i^2}{2})}{\sqrt{2\pi}} \cdot w_i^{\otimes k} \qquad (4)$$

We prove this by considering higher order derivatives and using properties of Hermite polynomials. A key property we use here is that the $k$'th derivative of a standard Gaussian function is itself multiplied by the $k$'th Hermite polynomial (with sign flipped for odd $k$). This significantly simplifies the expression for the coefficient $g(b_i)$ of $w_i^{\otimes k}$.

We remark that the above lemma may also be used to give an expression for the training objective for depth-2 ReLU networks, analogous to the result of [GLM18] for ReLU activations with no bias, that provides an expression as a combination of tensor decomposition problems of increasing order. Please refer to Appendix B for more details.

# 4 Non-robust Algorithm and Analysis

Our algorithms for learning the parameters of $f(x) = a^\top \sigma(W^\top x + b)$ decompose tensors obtained from the Hermite coefficients $\{\hat{f}_t \in (\mathbb{R}^d)^{\otimes t}\}$ of the function $f$. In this section, we design an algorithm assuming that we have access to all the necessary Hermite coefficients exactly (no noise or sampling errors). This will illustrate the basic algorithmic ideas and the identifiability result. However with polynomial samples, we can only hope to estimate these quantities up to inverse polynomial accuracy. In Section 5 we describe how we deal with the challenges that arise from errors. For proofs for the following lemmas and claims, please refer to Appendix C.

Our first result is a polynomial time algorithm in the full-rank setting that recovers all the parameters exactly.

**Theorem 4.1** (Full-rank non-robust setting). *Suppose the parameters $\{(a_i, b_i, w_i) : i \in [m]\}$ satisfies:* (i) *$a_i \neq 0$ for all $i \in [m]$,* (ii) *$\{w_i : i \in [m]\}$ are linearly independent. Then given $\{\hat{f}_t : 0 \leq t \leq 4\}$ exactly, Algorithm 1 recovers (with probability 1) the unknown parameters $a$, $b$ and $W$ in $\mathrm{poly}(m, d)$ time.*

See Theorem 3.1 for the analogous theorem in the presence of errors in estimating the Hermite coefficients $\{\hat{f}_t\}$. Our algorithm for recovering the parameters estimates different Hermite coefficient tensors $\{\hat{f}_t : 0 \leq t \leq 4\}$ and uses tensor decomposition algorithms on these tensors to first find the $\{w_i : i \in [m]\}$ up to some ambiguity in signs. We can also recover all the coefficients $\{b_i : i \in [m]\}$ up to signs (corresponding to the signs of $w_i$), and all the $\{a_i : i \in [m]\}$ (no sign ambiguities). This portion of the algorithm extends to higher order $\ell$, under a weaker assumption on the matrix $W$.

**Theorem 4.2.** *Suppose the parameters $\{(a_i, b_i, w_i) : i \in [m]\}$ satisfies:* (i) *no two $\{w_i : i \in [m]\}$ are linearly dependent and,* (ii) *for a constant $\ell \in \mathbb{N}$, $\{w_i^{\otimes \ell} : i \in [m]\}$ are linearly independent,* (iii) *$a_i \neq 0$ for all $i \in [m]$. Then given $\{\hat{f}_t : 0 \leq t \leq 2\ell + 2\}$ exactly, Algorithm 1 in $\mathrm{poly}_\ell(m, d)$ time outputs (with probability 1) $\{\hat{a}_i, \hat{w}_i, \hat{b}_i : i \in [m]\}$ such that we can recover the parameters up to a reordering of the indices $[m]$ and up to signs i.e., for some $\{\xi_i \in \{-1, 1\} : i \in [m]\}$ we have $\hat{a}_i = a_i$, $\hat{w}_i = \xi_i w_i$ and $\hat{b}_i = \xi_i b_i$. Furthermore, given exact statistical query access to the distribution $\mathcal{N}(0, I_{d \times d})$,[2] there exists an algorithm that runs in time $\mathrm{poly}(m, d, B)$ and outputs a function $g(x)$ such that $\mathbb{E}_{x \sim \mathcal{N}(0, I_{d \times d})}\big(f(x) - g(x)\big)^2 = 0$.*

We now describe the algorithm for general $\ell \geq 1$ (this specializes to the full-rank setting for $\ell = 1$).

---

**Algorithm 1:** Algorithm for order $\ell$: recover $a$, $b$, $W$ given $\{\hat{f}_t : 0 \leq t \leq 2\ell + 2\}$

---

**Input:** $\hat{f}_\ell, \hat{f}_{\ell+1}, \hat{f}_{\ell+2}, \hat{f}_{\ell+3}, \hat{f}_{2\ell+1}, \hat{f}_{2\ell+2}$;

1. Let $T' = \mathrm{flatten}(\hat{f}_{2\ell+1}, \ell, \ell, 1) \in \mathbb{R}^{d^\ell \times d^\ell \times d}$ and $T'' = \mathrm{flatten}(\hat{f}_{2\ell+2}, \ell, \ell, 2) \in \mathbb{R}^{d^\ell \times d^\ell \times d^2}$ be order-3 tensors obtained by flattening $\hat{f}_{2\ell+1}$ and $\hat{f}_{2\ell+2}$.

2. Set $k' = \mathrm{rank}(\mathrm{flatten}(\hat{f}_{2\ell+1}, \ell, \ell+1, 0))$. Run Jennrich's algorithm [Har70] on $T'$ to recover rank-1 terms $\{\alpha_i' u_i^{\otimes \ell} \otimes u_i^{\otimes \ell} \otimes u_i \mid i \in [k']\}$, where $\forall i \in [k']$, $u_i \in \mathbb{S}^{d-1}$ and $\alpha_i' \in \mathbb{R}$.

3. Set $k'' = \mathrm{rank}(\mathrm{flatten}(\hat{f}_{2\ell+2}, \ell, \ell+1, 0))$. Run Jennrich's algorithm [Har70] on $T''$ to recover rank-1 terms $\{\alpha_i'' v_i^{\otimes \ell} \otimes v_i^{\otimes \ell} \otimes v_i^{\otimes 2} \mid i \in [k'']\}$, where $\forall i \in [k'']$, $v_i \in \mathbb{S}^{d-1}$ and $\alpha_i'' \in \mathbb{R}$.

4. Remove duplicates and negations (i.e., antipodal pairs of the form $v$ and $-v$) from $\{u_1, u_2, \ldots, u_{k'}\} \cup \{v_1, v_2, \ldots, v_{k''}\}$ to get $\widetilde{w}_1, \widetilde{w}_2, \ldots, \widetilde{w}_m$.

5. Run subroutine RECOVERSCALARS$(m, \ell, \{\widetilde{w}_i : i \in [m]\}, \hat{f}_\ell, \hat{f}_{\ell+1}, \hat{f}_{\ell+2}, \hat{f}_{\ell+3})$ to get $\{\widetilde{a}_i, \widetilde{b}_i : i \in [m]\}$.

6. If $\ell = 1$ (full-rank setting), run Algorithm 3 (FIXSIGNS) on parameters $m, \hat{f}_1$ and $(\widetilde{a}_i, \widetilde{b}_i, \widetilde{w}_i : i \in [m])$ to get $(a_i' = \widetilde{a}_i, b_i', w_i' : i \in [m])$.

**Result:** Output $\{\widetilde{w}_i, \widetilde{a}_i, \widetilde{b}_i : i \in [m]\}$

---

---

[2]This means that for any function $h(x, y)$ that can be computed in polynomial time, one can obtain $\mathbb{E}_{x,y}[h(x, y)]$ exactly.

Subroutine Algorithm 2 finds the unknown parameters $a_1, \ldots, a_m \in \mathbb{R}$ and $b_1, \ldots, b_m \in \mathbb{R}$ given $w_1, w_2, \ldots, w_m$. While Algorithm 1 changes a little when we have errors in the estimates, the subroutine Algorithm 2 remains the same even for the robust version of the algorithm.

---

**Algorithm 2: Subroutine** RECOVERSCALARS to recover $b_i$ (up to signs) and $a_i$ given $w_i$ (up to signs) for all $i \in [m]$.

---

**Input:** $m, \ell, (\widetilde{w}_i : i \in [m])$, and tensors $T_\ell, T_{\ell+1}, T_{\ell+2}, T_{\ell+3}$ which are tensors of orders $\ell, \ell+1, \ell+2, \ell+3$ respectively;

**if** $\ell = 1$ **then**

    1. For $j \in \{2, 3\}$, solve the system of linear equation $\sum_{i=1}^m \zeta_j(i)\mathsf{vec}(\widetilde{w}_i^{\otimes j}) = \mathsf{vec}(T_j)$ to recover unknowns $\{\zeta_j(i) \mid i \in [m]\}$;

    2. For each $i \in [m]$, set $b_i = -\frac{\zeta_3(i)}{\zeta_2(i)}$, and $a_i = \zeta_2(i) \cdot \sqrt{2\pi}e^{b_i^2/2}$.

**else**

    1. For $j \in \{\ell, \ell+1, \ell+2, \ell+3\}$, solve the system of linear equation $\sum_{i=1}^m \zeta_j(i)\mathsf{vec}(\widetilde{w}_i^{\otimes j}) = \mathsf{vec}(T_j)$ to recover unknowns $\{\zeta_j(i) \mid i \in [m]\}$;

    2. For each $i \in [m]$, $q_i := \mathrm{argmax}_{j \in \{\ell+1, \ell+2\}} |\zeta_j(i)|$, set $\widetilde{b}_i = -\frac{\zeta_{q+1}(i) + q \cdot \zeta_{q-1}(i)}{\gamma_q(i)}$, and $\widetilde{a}_i = \sqrt{2\pi}(-1)^q \zeta_q(i) e^{b_i^2/2}/He_q(b_i)$, as described in (34) (Lemma 4.5).

**end**

**Result:** Output $(\widetilde{a}_i, \widetilde{b}_i : i \in [m])$

---

The above two algorithms together recover for all $i \in [m]$, the $a_i$ and up to a sign the $w_i$ and $b_i$. In the special case of $\ell = 1$ which we refer to as the *full-rank setting*, we can also recover the correct signs, and hence recover all the parameters.

---

**Algorithm 3: Algorithm** FIX SIGNS IN FULL-RANK SETTING.

---

**Input:** $m, \hat{f}_1 \in \mathbb{R}^d$ and estimates $\tilde{a}_i, \widetilde{b}_i, \tilde{w}_i$ for each $i \in [m]$;

1. Solve the system of linear equation $\sum_{i=1}^m z_i \tilde{a}_i \tilde{w}_i = \hat{f}_1$ to recover unknowns $\{z_i \mid i \in [m]\}$;

2. Set $\tilde{\xi}_i = \mathrm{sign}(z_i)$ for each $i \in [m]$.

**Result:** Output $(\tilde{a}_i, \tilde{\xi}_i \widetilde{b}_i, \tilde{\xi}_i \tilde{w}_i : i \in [m])$

---

Algorithm 1 decomposes two different tensors obtained from consecutive Hermite coefficients $\hat{f}_{2\ell+1}, \hat{f}_{2\ell+2}$ to obtain the $\{w_i\}$ up to signs. We use two different tensors because the bias $b_i$ could make the coefficient of the $i$th term in the decomposition 0 (e.g., $He_{2\ell-1}(b_i) = 0$ for $\hat{f}_{2\ell+1}$); hence $w_i$ cannot be recovered by decomposing $\hat{f}_{2\ell+1}$. Hence $\hat{f}_{2\ell+1}$ can degenerate to a rank $m' < m$ tensor, and Jennrich's algorithm will return only $m' < m$ eigenvectors that correspond to non-zero eigenvalues.

The following lemma addresses this issue by showing that two consecutive Hermite polynomials can not both take small values at any point $x \in \mathbb{R}$. This implies a separation between roots of consecutive Hermite polynomials or , and establishes a "robust" version that will be useful in Section 5. Moreover, this lemma also shows that when $|x|$ is not close to 0, at least one out of every two consecutive *odd* Hermite polynomials takes a value of large magnitude at $x$.

**Lemma 4.3** (Separation of Roots). *For all $k \in \mathbb{N}, x \in \mathbb{R}$, $\max\{|He_k(x)|, |He_{k+1}(x)|\} \geq \sqrt{k!/2}$.*

The following claim shows that Jennrich's algorithm for decomposing a tensor successfully recovers all the rank-1 terms whose appropriate $He_k(b_i) \neq 0$. This claim along with Lemma 4.3 shows that Steps 2-3 of Algorithm 1 successfully recovers all the $\{w_i : i \in [m]\}$ up to signs.

**Claim 4.4.** *Let $\ell_1, \ell_2 \geq \ell$ and $T \in \mathbb{R}^{d^{\ell_1} \times d^{\ell_2} \times d^{\ell_3}}$ have a decomposition $T = \sum_{i=1}^m \alpha_i w_i^{\otimes \ell_1} \otimes w_i^{\otimes \ell_2} \otimes w_i^{\otimes \ell_3}$, with $\{w_i^{\otimes \ell} : i \in [m]\}$ being linearly independent. Consider matrix $M = \textit{flatten}(T, \ell_1, \ell_2 + \ell_3, 0) \in \mathbb{R}^{d^{\ell_1} \times d^{\ell_2 + \ell_3}}$, and let $r := rank(M)$. Then Jennrich's algorithm applied with rank $r$ runs in $\mathrm{poly}_{\ell_1 + \ell_2 + \ell_3}(m, d)$ time recovers (w.p. 1) the rank-1 terms corresponding to $\{i \in [m] : |\alpha_i| > 0\}$. Moreover for each $i$ with $|\alpha_i| > 0$, we have $\tilde{w}_i = \xi_i w_i$ for some $\xi_i \in \{+1, -1\}$.*

The above claim was useful in recovering $w_i$ up to a sign ambiguity. The following lemma is useful for recovering $a_i$ parameters (no sign ambiguities) and the $b_i$ parameters up to sign ambiguity, once we have recovered the $w_i$ up to sign ambiguity. It uses various properties of Hermite polynomials along with Lemma 3.5 and Lemma 4.3.

**Lemma 4.5.** *Suppose* $k \in \mathbb{N}, k \geq 2$. *Suppose for some unknowns* $\beta, z \in \mathbb{R}$ *with* $\beta \neq 0$, *we are given values of* $\gamma_j = (-1)^j \xi^j \beta He_j(z) \, \forall j \in \{k, k+1, k+2, k+3\}$ *for some* $\xi \in \{+1, -1\}$. *Then* $z, \beta$ *are uniquely determined by*

$$\text{For } q := \underset{j \in \{k+1, k+2\}}{\mathrm{argmax}} |\gamma_j|, \;\; \xi z := -\frac{\gamma_{q+1} + q \cdot \gamma_{q-1}}{\gamma_q}, \quad \beta = (-1)^q \frac{\gamma_q}{He_q(\xi z)} \tag{5}$$

A robust version of this lemma (see Section D.2.2) will be important in the robust analysis of Section 5. The following claim applies the above lemma for each $i \in [m]$ with $\xi_i z = b_i$ and $\beta = a_i e^{-b_i^2/2}/\sqrt{2\pi}$, to show that Step 5 of the algorithm recovers the correct $\{(a_i, \xi_i b_i)\}_{i \in [m]}$ given the $\{\xi_i w_i\}_{i \in [m]}$.

**Claim 4.6.** *Given* $\{\widetilde{w}_i = \xi_i w_i : i \in [m]\}$ *where* $\xi_i \in \{+1, 1\} \, \forall i \in [m]$, *Step 5 of Alg. 1 recovers* $\{(a_i, \xi_i b_i, \xi_i w_i) : i \in [m]\}$.

We now complete the proof of the non-robust analysis for any constant $\ell \geq 0$.

*Proof of Theorem 4.2.* The proof follows by combining Claim 4.4 and Claim 4.6, along with Lemma 4.3. Let $Q_1 = \{i : |He_{2\ell-1}(b_i)| > 0\}$ and $Q_2 = \{i : |He_{2\ell+1}(b_i)| > 0\}$. From Claim 4.4, Step 2 recovers all the rank-1 terms in $Q_1$ with probability 1; hence we obtain in particular $\{\xi_i w_i \in \mathbb{S}^{d-1} \mid i \in Q_1\}$ for some signs $\xi_i \in \{+1, -1\}$. Similarly, in Step 3 we recover w.p. 1, the $\{\xi_i w_i \in \mathbb{S}^{d-1} \mid i \in Q_2\}$ for some signs $\xi_i \in \{1, -1\}$.

From Lemma 4.3, we know that no $x \in \mathbb{R}$ is a simultaneous root of $He_{2\ell+1}(x), He_{2\ell+2}(x)$. Hence $Q_1 \cup Q_2 = \{1, 2, \ldots, m\}$. Thus we obtain $\{\xi_i w_i \in \mathbb{S}^{d-1} : i \in [m]\}$ in Step 4 for some signs $\xi_i \in \{1, -1\}$ for all $i \in [m]$. Finally using Claim 4.6, we recover for each $i \in [m]$, the $a_i, \xi_i b_i \in \mathbb{R}$ corresponding to $\xi_i w_i$.

Next, in order to recover a function $g(x)$ of zero $L_2$ error we set up a linear regression problem. Given $x \in \mathbb{R}^d$ consider a $2m$ dimensional feature space $\phi(x)$ where $\phi(x)_{2i} = a_i \sigma(\xi_i w_i^\top x + \xi_i b_i)$ and $\phi(x)_{2i+1} = a_i \sigma(-\xi_i w_i^\top x - \xi_i b_i)$. Then is is easy to see that the target network $f(x)$ can be equivalently written as $f(x) = \beta^{*\top} \phi(x)$ for some vector $\beta^*$. Hence we can recover another vector $\beta$ of zero $L_2$ error by solving ordinary least squares, i.e, $\beta = \mathbb{E}[\phi(x)^\top \phi(x)]^\dagger \mathbb{E}[\phi(x)y]$.[3] Notice that both the expectations can be calculated exactly given exact statistical query access to the data distribution. In Section D.3 we provide a more general analysis of the above argument with finite sample analysis that will let us approximate $f(x)$ up to arbitrary accuracy in the presence of sampling errors. $\square$

We now complete the proof of recovery in the full-rank setting.

*Proof of Theorem 4.1.* We first apply Theorem 4.2 (and its above proof) with $\ell = 1$. We note that the conditions are satisfied since $s_{\min}(W) > 0$ and all the $a_i \neq 0$. Theorem 4.2 guarantees that the first 5 steps of Algorithm 1 recovers (with probability 1) for each $i \in [m]$, $a_i, \widetilde{b}_i = \xi_i b_i$ and $\widetilde{w}_i = \xi_i w_i$ for some $\xi_i \in \{+1, 1\}$. From Lemma 3.5, we have that

$$\hat{f}_1 = \sum_{i=1}^{m} a_i \Phi(b_i) w_i = \sum_{i=1}^{m} z_i^* a_i \widetilde{w}_i, \;\; \text{for } z_i^* = \xi_i \Phi(b_i) \, \forall i \in [m],$$

and $\Phi(b_i)$ is the Gaussian CDF and restricted to $(0, 1)$. Moreover the $\{w_i : i \in [m]\}$ are linearly independent. Hence there is a unique solution $(z_i^* : i \in [m])$ to the system of linear equations in the unknowns $(z_i : i \in [m])$ in step 6 of Algorithm 1, and $\widetilde{\xi}_i = \xi_i$ as required. Hence $(a_i, \widetilde{\xi}_i \widetilde{b}_i, \widetilde{\xi}_i \widetilde{w}_i : i \in [m])$ are the true parameters of the network (up to reordering indices). $\square$

---

[3] We remark that using Claim D.14, we can further consolidate the terms to get a ReLU network with at most $m + 2$ hidden units.

**Proof of Identifiability (Theorem 3.4):** Theorem 3.4 follows by verifying that the conditions of Theorem 4.2 hold for $\ell = m$. Conditions (i) and (iii) follow from the conditions of Theorem 3.4.

We now verify condition (ii). For a matrix $U$ with columns $\{u_i : i \in [m]\}$, the $\mathsf{krank}(U)$ (denoting the Kruskal-rank) is at least $k$ iff every $k$ of the $m$ columns of $U$ are linearly independent. Note that $\mathsf{krank}(U) \le \mathsf{rank}(U) \le m$. The krank increases under the Khatri-Rao product.

**Fact 4.7** (Lemma A.4 of [BCV14]). *For two matrices $U, V$ with $m$ columns, $\mathsf{krank}(U \odot V) = \min(\mathsf{krank}(U) + \mathsf{krank}(V) - 1, m)$.*

**Non-identifiability of signs when $\{w_i : i \in [m]\}$ are not linearly independent.** Theorem 4.1 shows that when the $\{w_i : i \in [m]\}$ are linearly independent, the model is identifiable. The following claim shows that even in the special setting when the biases $b_i = 0 \, \forall i \in [m]$, whenever the $(w_i : i \in [m])$ are linearly dependent, the model is non-identifiable (for appropriate $(a_i : i \in [m])$) because of ambiguities in the signs (Theorem 4.2 also shows it is identifiable up to this sign ambiguity).

**Claim 4.8.** *Suppose $w_1, \ldots, w_m$ are linearly dependent. Then there exists $a_1, \ldots, a_m$ (not all 0) and signs $\xi_1, \xi_2, \ldots, \xi_m \in \{\pm 1\}$ with not all $+1$ such that the ReLU networks $f$ and $g$ defined as*

$$f(x) := \sum_{i=1}^{m} a_i \sigma(w_i^\top x), \;\; g(x) := \sum_{i=1}^{m} a_i \sigma(\xi_i w_i^\top x) \text{ satisfy } f(x) = g(x) \, \forall x \in \mathbb{R}^d.$$

## 5   Robustness Analysis

In this section, we prove Theorem 3.1 and Theorem 3.2 which give polynomial time and sample complexity bounds for our algorithms. In the previous section we showed that given oracle access to $\{\hat{f}_k\}$, we can recover the exact network parameters $a, b, W$ (or at least up to signs). In reality, we can only access polynomially many samples in polynomial time, and we will have sampling errors when estimating $\{\hat{f}_k\}$. Therefore, given data generated from the target network $(x_1, y_1), ..., (x_N, y_N)$, we will approximate $\hat{f}_k$ through the empirical estimator

$$T_k = \frac{1}{N} \sum_{i=1}^{N} y_i He_k(x_i) \tag{6}$$

Observe that $T_k$ is an unbiased estimator for $\hat{f}_k$. We first show using standard concentration bounds that for any $\eta > 0$, with $N \ge \mathrm{poly}(d^k, m, B, 1/\eta)$ samples, the empirical estimates with high probability satisfies $\forall \ell \in [k]$, $\|\xi_\ell\|_F := \|T_\ell - \hat{f}_\ell\|_F \le \eta$ (see Appendix D.1). Hence for any constant $k$, with polynomial samples, we can obtain with high probability, estimates for the tensors $\{T_0, T_1, \ldots, T_k\}$ that are accurate up to any desired inverse-polynomial error.

**Overview of Analysis.** The error in the tensors $T_k$ introduces additional challenges that we described in Section 1. The analysis is technical and long, but we now briefly describe the main components.

*(i)* Recall from Section 1, that when there are errors, it may not even be possible to recover the parameters of some ReLU units! In particular when the bias $b_i$ is large in magnitude, the ReLU unit will be indistinguishable from a simple linear function. It will contribute negligibly to any of the higher order Hermite coefficients, and hence will be impossible to recover them individually (especially if there are multiple such units). For a desired recovery error $\varepsilon > 0$, the $m$ hidden ReLU units are split into groups (for analysis)

$$G = \{i \in [m] \mid |b_i| < O\big(\sqrt{\log(1/(\varepsilon \cdot mdB))}\big)\}, \text{ and } P = \{1, 2, \ldots, m\} \setminus G.$$

We aim to recover all of the parameters of the units corresponding to $G$ up to signs. For the terms in $P$, we will show the existence of a linear function that approximates the total contribution from all the terms in $P$.

*(ii)* The tensor decomposition steps (steps 2-3) are simpler in the no-noise setting: the parameter $w_i$ of the $i$th ReLU unit can be recovered (up to sign ambiguity) as long its bias $b_i$ is not a root of $He_{2\ell-1}$. When there is noise, there could be terms $i \in [m]$ for which $b_i$ are not roots of $He_{k-2}(x)$, and yet their signal can get swamped by the sampling error in the tensor. We can only hope to

recover those $k \leq m$ components whose corresponding coefficient is above some chosen threshold $\eta_1$ (the other terms are considered as part of the error tensor). However a technical issue that arises is that the robust recovery guarantees for tensor decomposition algorithms lose polynomial factors in different parameters including the least singular value ($s_k(\cdot)$) of the factor matrices. Hence, for each of step 2 and 3, we argue that recovery is possible only if the coefficient of the corresponding term is significantly large, and this may give reasonable estimates for only a subset of these $m$ terms with coefficients $> \varepsilon$.

*(iii)* When decomposing two consecutive tensors $T_{2\ell+1}$ and $T_{2\ell+2}$, we use Lemma 4.3 to argue that each $i \in G$ will have a large coefficient in at least one of these two tensors. Hence we can stitch together estimates $\{\widetilde{w}_i : i \in G\}$ which are accurate up to a sign and small error. This will in turn be used to recover $a_i, b_i$ for $i \in G$, with properties of Hermite polynomials used to ensure that the errors do not propagate badly.

*(iv)* We argue that the other ReLU units in $P = [m] \setminus G$ can be approximated altogether using a linear function. This is obtained by subtracting from estimates $T_0, T_1$ with the corresponding terms from $G$.

*(v)* The above arguments let us compute good approximations to the parameters for the units in $G$, but only up to signs. In order to use this to learn a good predictor for $f(x)$ we consider solving a truncated linear regression problem in an expanded feature space. At a high level, for each $i \in G$, given estimates $(\widetilde{a}_i, \widetilde{w}_i, \widetilde{b}_i)$ we consider an expanded feature representation for this unit into an 8-dimensional vector where each coordinate is of the form $\xi_{i_3} \widetilde{a}_i \sigma(\xi_{i_1} \widetilde{w}_i \cdot x + \xi_{i_2} \widetilde{b}_i)$ for $\xi_{i_1}, \xi_{i_2}, \xi_{i_3} \in \{-1, +1\}$.[4] Repeating this for every $i \in G$ it is easy to see that there is a linear function in the expanded space that approximates the part of the function $f(x)$ that depends on units in $G$. Combining with the previous argument that the units in $P = [m] \setminus G$ can be approximated by a linear function in the original feature space, we deduce that there is an $O(d + m)$ dimensional feature space where $f(x)$ admits a good linear approximation. We then solve a truncated least squares problem in this space to obtain our final function $g(x)$ that approximates $f(x)$ in $L_2$ error.

We will state the main claims and intermediate steps that prove the robustness of the algorithm and establish Theorems D.1, 3.1, 3.2 in Appendix D.

## 6 Conclusion

In this paper, we designed polynomial time algorithms for learning depth-2 neural networks with general ReLU activations (with non-zero bias terms), and gave provable guarantees under mild non-degeneracy conditions. The results of this work are theoretical in nature, in trying to understand whether efficient algorithms exist for learning ReLU networks; hence we believe they do not have any adverse societal impact. We addressed multiple challenges for learning such ReLU network with non-zero bias terms throughout our analyses, that may be more broadly useful in handling bias terms in the ReLU activations. We also proved identifiability under minimal assumptions and adopted the framework of smoothed analysis to establish beyond-worst-case guarantees. The major open direction is to provide similar guarantees for networks of higher depth.

## 7 Acknowledgement

The last two authors are supported by the National Science Foundation (NSF) under Grant No. CCF-1652491, CCF-1637585 and CCF-1934931.

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
