# A More Preliminaries

For some parameters $n_1, n_2, \dots$ we will say that a quantity $q \leq \text{poly}(n_1, n_2, \dots)$ if and only if there exists constants $c_0 > 0, c_1 > 0, c_2 > 0, \dots$ such that $q \leq c_0 n_1^{c_1} n_2^{c_2} \dots$. If these constant depend on another parameter $\ell > 0$ which is also a constant, then we will denote this by $\text{poly}_\ell(n_1, n_2, \dots)$. We will say that an probabilistic event occurs with high probability if and only if it occurs with probability $1 - (mdB)^{-\omega(1)}$ i.e., the failure probability is smaller that *any* inverse polynomial in the parameters $m, d, B$. Finally, we will use sign variables of the form $\xi_i \in \{1, -1\}$; they will typically capture an ambiguity in the sign of the parameters of the $i$th unit.

## A.1 Hermite Polynomials

**Definition A.1.** Let $D_t^{(k)}$ be the total differential operator taken $k$ times with respect to $t$. For a function $g : \mathbb{R}^d \to \mathbb{R}$, $D_t^{(k)} g(t)|_{t=t_0} \in (\mathbb{R}^d)^{\otimes k}$, where the $\alpha$'th element of $D_t^{(k)} g(t)$, for $\alpha = (i_1, i_2, \dots, i_k)$, is $\frac{d}{dt_{i_1}} \dots \frac{d}{dt_{i_k}} g(t)$ with $\alpha \in [d]^k$ being a multi-index. Note that the above is invariant to permutations, i.e., for any permutation $\alpha'$ of the indices in $\alpha$, the $\alpha'$th element of $D_t^{(k)} g(t)$ is the same as the $\alpha$th element.

**Definition A.2.** Let $x \in \mathbb{R}^d$, $\gamma(x) = \exp(-\frac{\|x\|^2}{2})$, the (probabilist's) $k$'th $d$-dimensional Hermite polynomial $He_k(x) \in (\mathbb{R}^d)^{\otimes k}$ is given by

$$He_k(x) = \frac{(-1)^k}{\gamma(x)} \cdot D_x^{(k)} \gamma(x) \tag{7}$$

A particularly useful fact for 1-dimensional Hermite polynomials is their relation with derivatives of a standard univariate Gaussian function.

**Fact A.3.** *The $k$'th order derivative of $\gamma(x)$ can be written in terms of 1-dimensional Hermite polynomials as*

$$\frac{d^k}{dx^k} \gamma(x) = (-1)^k \cdot He_k(x) \cdot \gamma(x) = He_k(-x)\gamma(x) \tag{8}$$

We will utilize this fact to express the Hermite coefficients of $f(x) = a^\mathsf{T} \sigma(W^\mathsf{T} x + b)$.

## A.2 Tensor Decomposition

The tensor product $u \otimes v \otimes w \in \mathbb{R}^{d_1} \otimes \mathbb{R}^{d_2} \otimes \mathbb{R}^{d_3}$ of vectors $u \in \mathbb{R}^{d_1}, v \in \mathbb{R}^{d_2}, w \in \mathbb{R}^{d_3}$ is a rank-1 tensor. Similarly, we will use $u^{\otimes \ell} \in (\mathbb{R}^d)^{\otimes \ell}$ to denote the tensor product of $u$ with itself $\ell$ times. An order-$t$ tensor $T \in \mathbb{R}^{d_1} \otimes \mathbb{R}^{d_2} \otimes \dots \otimes \mathbb{R}^{d_t}$ is represented using a $t$-way array $\mathbb{R}^{d_1 \times d_2 \times \dots \times d_t}$ that has $t$ modes corresponding to the $t$ different indices. Given two matrices $U, V$ with $k$ columns each given by $U = (u_i : i \in [k])$ and $V = (v_i : i \in [k])$, the *Khatri-Rao product* $M = U \odot V$ is a matrix formed by the $i$th column being $u_i \otimes v_i$. We will also use $U^{\odot 2} = U \odot U$ (and similarly for higher orders). A claim about preserving the full-column-rank property (and analogously minimum singular value) under the Khatri-Rao product is included below.

*Flattening or Reshaping:* Given an order-$t$ tensor $T$, for $t_1, t_2, t_3 \geq 0$ such that $t_1 + t_2 + t_3 = t$ define $T' = \text{flatten}(T, t_1, t_2, t_3)$ as the order-3 tensor $T' \in \mathbb{R}^{d^{t_1} \times d^{t_2} \times d^{t_3}}$, obtained by flattening and combining the first $t_1$ modes, the next $t_2$ and last $t_3$ modes respectively. When $t_3 = 0$, the output is a matrix in $\mathbb{R}^{d^{t_1} \times d^{t_2}}$.

Tensor decompositions of order 3 and above, unlike matrix decompositions (which are of order 2) are known to be unique under mild conditions. While tensor decompositions are NP-hard in the worst-case, polynomial time algorithms for tensor decompositions are known under certain non-degeneracy conditions (see e.g., [JGKA19, Vij20]). In particular, Jennrich's algorithm [Har70] provides polynomial time guarantees for recovering all the rank-1 terms of a decomposition of a tensor $T = \sum_{i=1}^k u_i \otimes v_i \otimes w_i$, when the $\{u_i : i \in [k]\}$ are linearly independent, the $\{v_i : i \in [k]\}$ are linearly independent, and no two of the vectors $\{w_i : i \in [k]\}$ are parallel. This algorithm and

its guarantee can also be made robust to some noise (of an inverse polynomial magnitude), when measured in Frobenius norm. In this paper, the following claims are especially vital to formulate our main results.

**Claim A.4** (Implication of Lemma A.4 of [BCV14][5]). *Let $U \in \mathbb{R}^{d_1 \times k}, V \in \mathbb{R}^{d_2 \times k}$, and suppose the smallest column length $\min_{i \in [k]} \|v_i\|_2 \geq \kappa$. Then the Khatri-Rao product $U \odot V$ has rank $k$ and satisfies $s_k(U \odot V) \geq \kappa \cdot s_k(U)/\sqrt{2k}$.*

A robust guarantee we will use for Jennrich's algorithm [Har70] is given below (see also [GVX14, Moi18] for robust analysis).

**Theorem A.5** (Theorem 2.3 of [BCMV14]). *Suppose $\varepsilon_{A.5} > 0$ we are given tensor $\widetilde{T} = T + E \in \mathbb{R}^{m \times n \times p}$, where $T$ has a decomposition $T = \sum_{i=1}^{k} u_i \otimes v_i \otimes w_i$ satisfying the following conditions:*

1. *Matrices $U = (u_i : i \in [k]), V = (v_i : i \in [k])$ have condition number (ratio of the maximum singular value $\sigma_1$ to the least singular value $\sigma_k$) at most $\kappa$,*

2. *For all $i \neq j$, the submatrix $W_{\{i,j\}}$ has $s_2(W_{\{i,j\}}) \geq \delta$.*

3. *Each entry of $E$ is bounded by $\eta_{A.5}(\varepsilon, \kappa, \max\{n, m, p\}, \delta) = \frac{\text{poly}(\varepsilon)}{\text{poly}(\kappa, \max\{n,m,p\}, \frac{1}{\delta})}$.*

*Then there exists a polynomial time algorithm that on input $\widetilde{T}$ returns a decomposition $\{(\widetilde{u}_i, \widetilde{v}_i, \widetilde{w}_i) : i \in [k]\}$ s.t. there is a permutation $p : [k] \to [k]$ with*

$$\forall i \in [k], \quad \|\widetilde{u}_i \otimes \widetilde{v}_i \otimes \widetilde{w}_i - u_{p(i)} \otimes v_{p(i)} \otimes w_{p(i)}\|_F \leq \varepsilon_{A.5}. \tag{9}$$

# B  Expressions for the Hermite Coefficients

**Lemma 3.5**  *The $k$'th Hermite expansion of $f(x) = a^\mathsf{T} \sigma(W^\top x + b)$, $\hat{f}_k$, when $k = 0, 1$, is*

$$\hat{f}_0 = \sum_{i=1}^{m} a_i \left[ b_i \Phi(b_i) + \frac{\exp(-\frac{b_i^2}{2})}{\sqrt{2\pi}} \right], \ \hat{f}_1 = \sum_{i=1}^{m} a_i \Phi(b_i) w_i \tag{10}$$

*when $k \geq 2$, the coefficients are*

$$\hat{f}_k = \sum_{i=1}^{m} (-1)^k \cdot a_i \cdot He_{k-2}(b_i) \cdot \frac{\exp(\frac{-b_i^2}{2})}{\sqrt{2\pi}} \cdot w_i^{\otimes k} \tag{11}$$

*where the expectation is taken over $x \sim \mathcal{N}(0, I_d)$ and $\hat{f}_k$ is a $k$'th-order tensor.*

*Proof.*  Note that Hermite polynomials can be written in terms of their generating function [O'D14]

$$He_k(x) = D_t^{(k)} \exp(t^\mathsf{T} x - \|t\|^2/2)|_{t=\mathbf{0}} \tag{12}$$

Hence we can write $\hat{f}_k$ as

$$\hat{f}_k = \mathbb{E}[f(x)He_k(x)] = \sum_{i=1}^{m} a_i \int_{w_i^\top x + b_i \geq 0} (w_i^\mathsf{T} x + b_i) \cdot D_t^{(k)} \exp(t^\mathsf{T} x - \frac{\|t\|^2}{2}) \, d\mu|_{t=\mathbf{0}} \tag{13}$$

where $d\mu = \frac{\exp(-\|x\|^2/2)}{(\sqrt{2\pi})^d} dx$ is the Gaussian probability measure. Moving $d\mu$ into the exponential term, we get

$$\hat{f}_k = \sum_{i=1}^{m} \frac{a_i}{(\sqrt{2\pi})^d} \int_{w_i^\top x + b_i \geq 0} (w_i^\mathsf{T} x + b_i) \cdot D_t^{(k)} \exp(-\frac{\|x\|^2}{2} + t^\mathsf{T} x - \frac{\|t\|^2}{2}) \, dx|_{t=\mathbf{0}} \tag{14}$$

---

[5]In fact one can show that a stronger statement that a certain quantity called Kruskal-rank increases, see [BCV14].

$$= \sum_{i=1}^{m} \frac{a_i}{(\sqrt{2\pi})^d} \int_{w_i^\top x + b_i \geq 0} (w_i^\top x + b_i) \cdot D_t^{(k)} \exp(-\frac{\|x - t\|^2}{2}) \, dx|_{t=\mathbf{0}} \tag{15}$$

Since $(w_i^\top x + b_i) \exp(-\frac{\|x - t\|^2}{2})$ is continuous in all $x \in \mathbb{R}^d$ and $t \in \mathbb{R}^d$, by Leibniz's integral rule we can take the differential operator out of the integral, yielding

$$= D_t^{(k)} \Big[ \sum_{i=1}^{m} \frac{a_i}{(\sqrt{2\pi})^d} \int_{w_i^\top x + b_i \geq 0} (w_i^\top x + b_i) \cdot \exp(-\frac{\|x - t\|^2}{2}) \, dx \Big]_{t=\mathbf{0}} \tag{16}$$

Denote

$$I_i(t) = \frac{1}{(\sqrt{2\pi})^d} \int_{w_i^\top x + b_i \geq 0} (w_i^\top x + b_i) \cdot \exp(-\frac{\|x - t\|^2}{2}) \, dx \tag{17}$$

then

$$\hat{f}_k = D_t^{(k)} \sum_{i=1}^{m} a_i I_i(t)|_{t=\mathbf{0}} \tag{18}$$

Now, let $y_i = w_i^\top x$ and $t_{w_i} = t^\top w_i$. To evaluate $I_i(t)$ in terms of $y_i$, it suffices to only consider the projection of $t$ onto $w_i$, $t_{w_i}$ with the remaining parts being integrated out. Hence, we can rewrite $I_i(t)$ as

$$I_i(t) = \frac{1}{(\sqrt{2\pi})^d} \int_{x' \in \mathbb{R}^{d-1}} \exp(-\frac{\|x' - t'\|^2}{2}) dx' \int_{y_i = -b_i}^{\infty} (y_i + b_i) \cdot \exp(-\frac{|y_i - t_{w_i}|^2}{2}) dy_i \tag{19}$$

$$= \frac{1}{\sqrt{2\pi}} \int_{y_i = -b_i}^{\infty} (y_i + b_i) \cdot \exp(-\frac{|y_i - t_{w_i}|^2}{2}) dy_i \tag{20}$$

$$= (t_{w_i} + b_i)\Phi(t_{w_i} + b_i) + \frac{\exp(-\frac{(t_{w_i} + b_i)^2}{2})}{\sqrt{2\pi}} \tag{21}$$

where $\Phi(z)$ is the standard Gaussian c.d.f. of $z$. We then have

$$\hat{f}_k = \sum_{i=1}^{m} a_i \cdot D_t^{(k)} \Big[ (t_{w_i} + b_i)\Phi(t_{w_i} + b_i) + \frac{\exp(-\frac{(t_{w_i} + b_i)^2}{2})}{\sqrt{2\pi}} \Big]_{t=\mathbf{0}} \tag{22}$$

Therefore we have $\hat{f}_0$ and $\hat{f}_1$ as

$$\hat{f}_0 = \sum_{i=1}^{m} a_i \Big[ b_i \Phi(b_i) + \frac{\exp(-\frac{b_i^2}{2})}{\sqrt{2\pi}} \Big] \, , \, \hat{f}_1 = \sum_{i=1}^{m} a_i \Phi(b_i) w_i \tag{23}$$

Since we are taking the derivative with respect to $t$, for some function $g(t_{w_i})$, by the chain rule we will have

$$D_t^{(k)} g = \frac{d^k g}{dt_{w_i}^k} \cdot w_i^{\otimes k} \tag{24}$$

Finally, recall Fact A.3, the derivatives of a Gaussian p.d.f. can be expressed in terms of Hermite polynomials, hence for $k \geq 2$

$$\hat{f}_k = \sum_{i=1}^{m} a_i \cdot \frac{d^{k-2}}{dt_{w_i}^{k-2}} \frac{\exp(-\frac{(b_i + t_{w_i})^2}{2})}{\sqrt{2\pi}} \cdot w_i^{\otimes k}|_{t_{w_i}=0} \tag{25}$$

$$= \sum_{i=1}^{m} (-1)^{k-2} \cdot a_i \cdot He_{k-2}(b_i + t_{w_i}) \cdot \frac{\exp(-\frac{(b_i + t_{w_i})^2}{2})}{\sqrt{2\pi}} \cdot w_i^{\otimes k}|_{t_{w_i}=0} \tag{26}$$

$$= \sum_{i=1}^{m} (-1)^k \cdot a_i \cdot He_{k-2}(b_i) \cdot \frac{\exp(\frac{-b_i^2}{2})}{\sqrt{2\pi}} \cdot w_i^{\otimes k} \tag{27}$$

which proves the lemma. $\square$

**Proposition B.1.** *Let $\widetilde{f}(x) = \widetilde{a}^\top \sigma(\widetilde{W}^\top x + \widetilde{b})$ be the model trained using samples generated by the ground-truth ReLU network $f(x) = a^\top \sigma(W^\top x + b)$. Then the statistical risk with respect to the $\ell_2$ loss function can be expressed as follows*

$$L(\widetilde{a}, \widetilde{b}, \widetilde{W}) = \sum_{k \in \mathbb{N}} \frac{1}{k!} \left\| T_k - \hat{f}_k \right\|_F^2$$

$$\text{where} \quad T_0 = \sum_{i=1}^m \widetilde{a}_i(\widetilde{b}_i \Phi(\widetilde{b}_i) + \frac{\exp(-\widetilde{b}_i^2/2)}{\sqrt{2\pi}}), \quad \text{and} \quad T_1 = \sum_{i=1}^m \widetilde{a}_i \Phi(\widetilde{b}_i) \widetilde{w}_i$$

$$\forall k \geq 2, \quad T_k = \sum_{i=1}^m (-1)^k \cdot \widetilde{a}_i \cdot He_{k-2}(\widetilde{b}_i) \cdot \frac{\exp(-\widetilde{b}_i^2/2)}{\sqrt{2\pi}} \cdot \widetilde{w}_i^{\otimes k}$$

*Proof.* Let $\alpha \bowtie \alpha'$ denote $\alpha$ being a permutation of $\alpha'$. Since $\bowtie$ is an equivalence relation, we can partition $[d]^*$ into equivalence classes ($*$ is the Kleene star operator) such that for some $\alpha \in [d]^*$, $[\alpha] = \{\alpha' \in [d]^* \mid \alpha \bowtie \alpha'\}$. Let $C$ be a subset of $[d]^*$ such that no pair of $\alpha, \alpha' \in C$ is in the same equivalence class. We can then directly decompose the statistical risk as

$$\mathbb{E}\Big[|\widetilde{f}(x) - f(x)|^2\Big] = \mathbb{E}\Big[\Big|\sum_{\alpha \in C} \frac{T_\alpha He_\alpha(x)}{c_\alpha^2} - \sum_{\alpha \in C} \frac{\hat{f}_\alpha He_\alpha(x)}{c_\alpha^2}\Big|^2\Big] \tag{28}$$

where $c_\alpha^2 = \mathbb{E}[He_\alpha(x)^2]$. Note that we omit $k$ here and directly write the Hermite polynomial obtained by differentiating with respect to $x_{\alpha_1}, ..., x_{\alpha_k}$ as $He_\alpha(x) \in \mathbb{R}$. The above equation can thus be further simplified as

$$\mathbb{E}\Big[\sum_{\alpha \in C} \Big(\frac{(T_\alpha - \hat{f}_\alpha)He_\alpha(x)}{c_\alpha^2}\Big)^2 + \sum_{\substack{\alpha \neq \alpha' \\ \alpha, \alpha' \in C}} \Big(\frac{(T_\alpha - \hat{f}_\alpha)He_\alpha(x)}{c_\alpha^2}\Big)\Big(\frac{(T_{\alpha'} - \hat{f}_{\alpha'})He_{\alpha'}(x)}{c_{\alpha'}^2}\Big)\Big] \tag{29}$$

$$= \sum_{\alpha \in C} \frac{(T_\alpha - \hat{f}_\alpha)^2}{c_\alpha^2} \mathbb{E}\Big[\frac{He_\alpha(x)^2}{c_\alpha^2}\Big] = \sum_{\alpha \in C} \frac{(T_\alpha - \hat{f}_\alpha)^2}{c_\alpha^2} \tag{30}$$

since if both $\alpha, \alpha' \in C$ and $\alpha \neq \alpha'$, $\mathbb{E}[He_\alpha(x)He_{\alpha'}(x)] = 0$. Next, we rewrite the expression as

$$\sum_{k \in \mathbb{N}} \sum_{\substack{\alpha \in C \\ |\alpha|=k}} \frac{(T_\alpha - \hat{f}_\alpha)^2}{c_\alpha^2} = \sum_{k \in \mathbb{N}} \frac{1}{k!} \sum_{\substack{\alpha \in C \\ |\alpha|=k}} \frac{k!}{c_\alpha^2}(T_\alpha - \hat{f}_\alpha)^2 = \sum_{k \in \mathbb{N}} \frac{1}{k!} \left\| T_k - \hat{f}_k \right\|_F^2 \tag{31}$$

The last equality is due to the fact that $c_\alpha^2 = \prod_{i=1}^d n_i!$, where $n_i$ is the number of times that $i$ occurs in the multi-index $\alpha$, and therefore $k!/c_\alpha^2$ is the number of possible permutations of the elements in $\alpha$ with $|\alpha| = k$ subjecting to 1 occurs $n_1$ times, 2 occurs $n_2$ times, ..., $d$ occurs $n_d$ times. Thus the proposition follows. $\square$

## C Proofs in Section 4

**Lemma 4.3** (Separation of Roots) *For all $k \in \mathbb{N}$, $x \in \mathbb{R}$, $\max\{|He_k(x)|, |He_{k+1}(x)|\} \geq \sqrt{k!/2}$.*

*Proof.* First, $\forall k \in \mathbb{N}$, by Turán's inequality [Tur50] we have

$$He_{k+1}^2(x) - He_k(x)He_{k+2}(x) = k! \cdot \sum_{i=0}^k \frac{He_i(x)^2}{i!} > 0 \tag{32}$$

Set $\varepsilon = \sqrt{k!/2}$ and assume for contradiction that $|He_{k+1}(x)|, |He_{k+2}(x)| < \varepsilon$. The LHS of (32) is at most $\varepsilon^2 + \varepsilon|He_k(x)|$, and the RHS of (32) is at least

$$k! \cdot (1 + x^2 + ... + \frac{He_k(x)^2}{k!}) > k! \cdot (1 + \frac{He_k(x)^2}{k!}) \tag{33}$$

Therefore, if $|He_k(x)| = t$, combining both sides we get $\varepsilon^2 + \varepsilon t \geq k! + t^2$. This implies on the one hand that $\varepsilon t \geq k!/2$, and on the other hand that $\varepsilon t \geq t^2$. However for our choices of $\varepsilon = \sqrt{k!/2}$, no value of $t$ is feasible. This yields the required contradiction for the first claim.

$\square$

**Claim 4.4** *Let* $\ell_1, \ell_2 \geq \ell$ *and* $T \in \mathbb{R}^{d^{\ell_1} \times d^{\ell_2} \times d^{\ell_3}}$ *have a decomposition* $T = \sum_{i=1}^{m} \alpha_i w_i^{\otimes \ell_1} \otimes w_i^{\otimes \ell_2} \otimes w_i^{\otimes \ell_3}$, *with* $\{w_i^{\otimes \ell} : i \in [m]\}$ *being linearly independent. Consider matrix* $M = $ *flatten*$(T, \ell_1, \ell_2 + \ell_3, 0) \in \mathbb{R}^{d^{\ell_1} \times d^{\ell_2 + \ell_3}}$, *and let* $r := rank(M)$. *Then Jennrich's algorithm applied with rank* $r$ *runs in* $\text{poly}_{\ell_1 + \ell_2 + \ell_3}(m, d)$ *time recovers (w.p. 1) the rank-1 terms corresponding to* $\{i \in [m] : |\alpha_i| > 0\}$. *Moreover for each* $i$ *with* $|\alpha_i| > 0$, *we have* $\tilde{w}_i = \xi_i w_i$ *for some* $\xi_i \in \{+1, -1\}$.

*Proof.* Let $Q = \{i \in [m] : |\alpha_i| > 0\}$. Firstly $r = rank(M) = |Q|$, since $M$ has a decomposition

$$M = \sum_{i \in Q} \alpha_i \left( w_i^{\otimes \ell_1} \right) \left( w_i^{\otimes \ell_2 + \ell_3} \right)^\top = M_1 \text{diag}(\alpha_Q) M_2^\top$$

where $M_1, M_2, \text{diag}(\alpha_Q)$ all have full column rank $|Q|$. Secondly, from assumption (ii) of Theorem 4.2 and Claim A.4 applied with $\ell_1, \ell_2 \geq \ell$, we have that $\{w_i^{\otimes \ell'} : i \in [m]\}$ are linearly independent for every $\ell' \geq \ell$. Hence the the factor matrices $U = (w_i^{\otimes \ell_1} : i \in Q)$ and $V = (w_i^{\otimes \ell_2} : i \in Q)$ also have full column rank. Similarly from (iii) no two vectors in $\{\alpha_i w_i^{\otimes \ell_3} : i \in Q\}$ are parallel. Hence, they satisfy the conditions of Jennrich's algorithm. Since there is no error in the tensor, Jennrich's algorithm (Theorem A.5) succeeds with probability 1 (see [Vij20]). Finally since each rank-1 term is recovered exactly when $\alpha_i \neq 0$, the vector in $\mathbb{R}^d$ obtained from the term will correspond to either $w_i$ or $-w_i$ as required. $\square$

**Lemma 4.5** *Suppose* $k \in \mathbb{N}, k \geq 2$. *Suppose for some unknowns* $\beta, z \in \mathbb{R}$ *with* $\beta \neq 0$, *we are given values of* $\gamma_j = (-1)^j \xi^j \beta He_j(z) \ \forall j \in \{k, k+1, k+2, k+3\}$ *for some* $\xi \in \{+1, -1\}$. *Then* $z, \beta$ *are uniquely determined by*

$$For \ q := \underset{j \in \{k+1, k+2\}}{\operatorname{argmax}} |\gamma_j|, \quad \xi z = -\frac{\gamma_{q+1} + q \cdot \gamma_{q-1}}{\gamma_q}, \quad \beta = (-1)^q \frac{\gamma_q}{He_q(\xi z)} \tag{34}$$

*Proof.* We use the following fact about Hermite polynomials:

$$He_{r+1}(z) = z He_r(z) - r \cdot He_{r-1}(z). \tag{35}$$

From Lemma 4.3, we know that $\max\{|He_{k+1}(z)|, |He_{k+2}(z)|\} > 0$ and hence $\gamma_q \neq 0$. Substituting in the recurrence (35) with $r = q$,

$$z = \frac{He_{q+1}(z) + q \cdot He_{q-1}(z)}{He_q(z)} = \frac{\beta He_{q+1}(z) + q \cdot \beta He_{q-1}(z)}{\beta He_q(z)} = -\xi \left( \frac{\gamma_{q+1} + q \cdot \gamma_{q-1}}{\gamma_q} \right),$$

where we used the fact that the Hermite polynomials are odd functions for odd $q$ and even polynomials for even $q$. The $\beta$ value is also recovered since $\gamma_q = (-1)^q \beta (\xi^q He_q(z)) = (-1)^q \beta \cdot He_q(\xi z)$.

$\square$

**Claim 4.6** *Given* $\{\tilde{w}_i = \xi_i w_i : i \in [m]\}$ *where* $\xi_i \in \{+1, 1\} \ \forall i \in [m]$, *Step 5 of Alg. 1 recovers* $\{(a_i, \xi_i b_i, \xi_i w_i) : i \in [m]\}$.

*Proof.* We first prove for $\ell \geq 2$. For each of the $j \in \{\ell, \ell+1, \ell+2, \ell+2\}$, we have from Lemma 3.5 and the Hermite polynomials $He_j$ being odd functions for odd $j$ and even functions for even $j$,

$$\hat{f}_j = \sum_{i=1}^{m} (-1)^j \cdot a_i \cdot \cdot \frac{He_{j-2}(b_i) \exp(-b_i^2/2)}{\sqrt{2\pi}} \cdot w_i^{\otimes j} = \sum_{i=1}^{m} (-1)^j \cdot a_i \cdot \frac{He_{j-2}(\xi_i b_i) \exp(-b_i^2/2)}{\sqrt{2\pi}} \cdot \tilde{w}_i^{\otimes j}.$$

Moreover the vectors $\{\xi_i^j w_i^{\otimes j} : i \in [m]\}$ are linearly independent by assumption (and from Claim A.4 for $j > \ell$). Hence the linear system for each $j$ has a unique solution

$$\forall \ell \leq j \leq \ell + 3, \ \forall i \in [m], \ \text{ we have } \zeta_j(i) = a_i \cdot \frac{(-1)^j}{\sqrt{2\pi}} \exp(-b_i^2/2) \, He_{j-2}(\xi_i b_i).$$

Lemma 4.5 applied with $\beta = \frac{1}{\sqrt{2\pi}} a_i e^{-b_i^2/2}$ and $\gamma_{j-2} = (-1)^j He_{j-2}(\xi_i b_i)$ (note $\beta \neq 0$) proves that Alg. 2 recovers $a_i, \xi_i b_i$.

For $\ell = 1$, we note $\forall z \in \mathbb{R}$, $He_0(z) = 1$, and $He_1(z) = z$. From Lemma 3.5, we see that one set of solutions to the linear system is

$$\forall i \in [m], \ \zeta_2(i) = a_i \frac{\exp(-b_i^2/2)}{\sqrt{2\pi}}, \quad \text{and} \quad \zeta_3(i) = -a_i \xi_i b_i \cdot \frac{\exp(-b_i^2/2)}{\sqrt{2\pi}}.$$

Moreover the vectors $\{\xi_i w_i : i \in [m]\}$ are linearly independent. Hence, $\zeta_2, \zeta_3$ are the unique solutions to the system. Hence Algorithm 2 recovers $a_i, \xi_i b_i$ as claimed. $\qquad\square$

**Fact 4.7** (Lemma A.4 of [BCMV14]) *For two matrices $U, V$ with $m$ columns,* $\mathsf{krank}(U \odot V) = \min(\mathsf{krank}(U) + \mathsf{krank}(V) - 1, m)$.

*Proof.* Let $U = W^\top$ (with $i$th column $w_i$). Note that since no two columns are parallel, $\mathsf{krank}(U) \geq 2$. By applying the above fact on matrix $M = U^{\odot m}$ with $i$th column $w_i^{\otimes m}$, we get that $\mathsf{krank}(M) = m$, as required. Hence, Theorem 4.2 can be applied to recover for all $i \in [m]$, all the unknown $a_i$, and up to ambiguities in signs given by (unknown) $\xi_i \in \{1, -1\}$ the $b_i$ and $w_i$ as well (we recover $\xi_i b_i, \xi_i w_i$).

For the second half of the claim, let $\xi_i \in \{1, -1\} \ \forall i \in [m]$ be any combination of signs. Consider the solution $a_i' = a_i, w_i' = \xi_i w_i, b_i' = \xi_i b_i$, and let $g(x)$ represent the corresponding ReLU function given by these parameters. Note that the Hermite polynomial $He_t(\xi z) = \xi^t He_t(z)$ for all $\xi \in \{\pm 1\}$ and $z \in \mathbb{R}$. Hence, the Hermite coefficients of order at least 2 are equal for $f$ and $g$ i.e., for all $t \geq 2$

$$\hat{g}_t = \sum_{i=1}^m (-1)^t a_i \, He_{t-2}(\xi_i b_i) \cdot \frac{e^{-b_i^2/2}}{\sqrt{2\pi}} \cdot (\xi_i w_i)^{\otimes t} = \sum_{i=1}^m (-1)^t a_i \, He_{t-2}(b_i) \cdot \frac{e^{-b_i^2/2}}{\sqrt{2\pi}} \cdot w_i^{\otimes t} = \hat{f}_t.$$

Condition 2 also implies that zeroth and first Hermite coefficients of $f, g$ are also equal. All the Hermite coefficients are hence equal (and the functions are squared-integrable w.r.t. the Gaussian measure for bounded $a_i$). Thus the two functions $f$ and $g$ being identical follows since Hermite polynomials form a complete orthogonal system. $\qquad\square$

**Claim 4.8** *Suppose $w_1, \ldots, w_m$ are linearly dependent. Then there exists $a_1, \ldots, a_m$ (not all $0$) and signs $\xi_1, \xi_2, \ldots, \xi_m \in \{\pm 1\}$ with not all $+1$ such that the ReLU networks $f$ and $g$ defined as*

$$f(x) \coloneqq \sum_{i=1}^m a_i \sigma(w_i^\top x), \ g(x) \coloneqq \sum_{i=1}^m a_i \sigma(\xi_i w_i^\top x) \text{ satisfy } f(x) = g(x) \ \forall x \in \mathbb{R}^d.$$

*Proof.* Since $\{w_i : i \in [m]\}$ are linearly dependent, there exists $(\beta_i : i \in [m])$ which are not all $0$ such that $\sum_{i=1}^n \beta_i w_i = 0$. Define $a_i = \beta_i$ for each $i \in [m]$, and let $\xi_i = -1$ if $\beta_i \neq 0$ and $\xi_i = 1$ otherwise. Let $f(x) = \sum_{i=1}^m a_i \sigma(w_i^\top x)$ and $g(x) = \sum_{i=1}^m a_i \sigma(\xi_i w_i^\top x)$.

From Lemma 3.5, it is easy to verify that all the even Hermite coefficients are equal, and the odd Hermite coefficients for $\ell \geq 3$ are all $0$ since $b_i = 0$. Moreover the $\ell = 1$ order Hermite coefficients are equal since

$$\sum_{i=1}^m a_i w_i - \sum_{i=1}^m a_i \xi_i w_i = \sum_{i=1}^m a_i (1 - \xi_i) w_i = \sum_{i=1}^m 2\beta_i w_i = 0.$$

All the Hermite coefficients of $f$ and $g$ are equal (and the functions are also squared-integrable w.r.t. the Gaussian measure when the $a_i$ are bounded). As the Hermite polynomials form a complete orthogonal basis, the two ReLU network functions $f(x)$ and $g(x)$ are also equal. This concludes the proof. $\qquad\square$

# D    Robust Analysis for general $\ell$

The main algorithm in the robust setting is Algorithm 4 described below, which approximately recovers the parameters for the activation units (up to signs) that do not have large positive bias. The guarantees are given in the following Theorem D.1.

**Theorem D.1.** *Suppose $\ell \in \mathbb{N}$ be a constant, and $\varepsilon > 0$. If we are given $N$ i.i.d. samples as described above from a ReLU network $f(x) = a^\top \sigma(W^\top x + b)$ that is $B$-bounded Then there are constants $c = c(\ell) > 0, c' > 0$, signs $\xi_i \in \{\pm 1\} \ \forall i \in [m]$ and a permutation $\pi : [m] \to [m]$ such that Algorithm 4 given $N \geq \operatorname{poly}_\ell(m, d, 1/\varepsilon, 1/s_m(W^{\odot \ell}), B)$ runs in $\operatorname{poly}_\ell(N, m, d)$ time and with high probability outputs $\{\widetilde{a}_i, \widetilde{b}_i, \widetilde{w}_i : i \in [m']\}$ such that for all $i \in [m]$ with $|b_i| < c\sqrt{\log(1/(\varepsilon \cdot mdB))}$ we have that $\|w_i - \xi_{\pi(i)}\widetilde{w}_{\pi(i)}\|_2 + |a_i - \widetilde{a}_{\pi(i)}| + |b_i - \xi_{\pi(i)}\widetilde{b}_{\pi(i)}| \leq \varepsilon$.*

In fact, the analysis just assumes that $\|T_k - \hat{f}_k\|_F$ are upper bounded up to an amount that is inverse polynomial in the different parameters (this could also include other sources of error) to approximate the 2-layer ReLU network that approximates $f$ up to desired inverse polynomial error $\varepsilon$.

---

**Algorithm 4:** for order $\ell$:  recover $\{a_i\}$, and (up to signs) $\{b_i, w_i\}$ given estimates $\{T_0, \ldots, T_{2\ell+2}\}$.

---

**Input:** Estimates $T_0, \ldots, T_{2\ell+2}$ for $\hat{f}_0, \hat{f}_1, \ldots, \hat{f}_{2\ell+2}$;
**Parameters:** $\eta_0, \eta_1, \eta_2, \eta_3 > 0$.;
1. Let order-3 tensors $T' = \operatorname{flatten}(T_{2\ell+1}, \ell, \ell, 1) \in \mathbb{R}^{d^\ell \times d^\ell \times d}$ and let
$T'' = \operatorname{flatten}(T_{2\ell+2}, \ell, \ell, 2) \in \mathbb{R}^{d^\ell \times d^\ell \times d^2}$.
2. Set $k' = \max_{r \leq m} s_r(\operatorname{flatten}(T_{2\ell+1}, \ell, \ell+1, 0)) > \eta_1$. Run Jennrich's algorithm on $T'$ to recover rank-1 terms $\{\alpha_i' u_i^{\otimes \ell} \otimes u_i^{\otimes \ell} \otimes u_i \mid i \in [k']\}$, where $\forall i \in [k']$, $u_i \in \mathbb{S}^{d-1}$ and $\alpha_i' \in \mathbb{R}$.
3. Set $k'' = \max_{r \leq m} s_r(\operatorname{flatten}(T_{2\ell+2}, \ell, \ell+2, 0)) > \eta_1$. Run Jennrich's algorithm on $T''$ to recover rank-1 terms $\{\alpha_i'' v_i^{\otimes \ell} \otimes v_i^{\otimes \ell} \otimes v_i^{\otimes 2} \mid i \in [k'']\}$, where $\forall i \in [k'']$, $v_i \in \mathbb{S}^{d-1}$ and $\alpha_i'' \in \mathbb{R}$.
4. Remove all the rank-1 terms in steps 2 and 3 with Frobenius norm $< \eta_2$ i.e., $\alpha_i''$ or $\alpha_i' < \eta_2$. Also remove all duplicates from $\{u_1, u_2, \ldots, u_{k'}\} \cup \{v_1, v_2, \ldots, v_{k''}\}$ even up to signs i.e., remove iteratively from the above set vectors $v$ if either of $+v, -v$ are within $\eta_3$ in $\ell_2$ distance of the other vectors in the set, to get $\widetilde{w}_1, \widetilde{w}_2, \ldots, \widetilde{w}_{m'}$.
5. Run the subroutine RECOVERSCALARS$(\ell, \{\widetilde{w}_i : i \in [m']\}, T_\ell, T_{\ell+1}, T_{\ell+2}, T_{\ell+3})$ (i.e., Alg. 2) to get $\{\widetilde{a}_i, \widetilde{b}_i : i \in [m']\}$.
**Result:** Output $\{\widetilde{w}_i, \widetilde{a}_i, \widetilde{b}_i : 1 \leq i \leq m'\}$.

---

The following algorithm (Algorithm 5) shows how to find a depth-2 ReLU network that fits the data i.e., achieves arbitrarily small $L_2$ error. The algorithm uses Algorithm 4 as a black-box to first approximately recover the unknown parameters of the activation units (with not very large bias) up to signs, and then setup an appropriate linear regression problem to find a network that fits the data.

The error parameters $\eta_0, \eta_1, \eta_2, \eta_3$ can be set with appropriate polynomial dependencies on $\varepsilon, d^\ell, m, B, s_m(W^{\odot \ell})$ to obtain the recovery guarantees in Theorem D.1 and Theorem 3.2.

In this section, we prove Algorithm 4 and its algorithmic guarantee in Theorem D.1 (and Theorems 3.1 and 3.2).

We break down the proof into multiple parts.

## D.1    Estimating the Hermite coefficients

First, we derive concentration bounds on $\xi_k$, which will be followed by error bounds of the recovered parameters $\widetilde{a}, \widetilde{b}, \widetilde{W}$ in terms of $\xi_k$. To obtain the desired concentration bound, we first introduce an auxiliary claim we will make use of in the following analysis.

**Claim D.2.** *For $a_1, a_2, \ldots, a_n \in \mathbb{R}$ and $p \in \mathbb{N}$, $|\sum_{i \in [n]} a_i|^{2p} \leq n^{2p} \max_{i \in [n]} |a_i|^{2p} \leq n^{2p} \sum_{i \in [n]} |a_i|^{2p}$*

**Algorithm 5:** Outputs a function $g(x)$ that approximates the target network $f(x)$ in mean squared error.

**Input:** $N$ i.i.d. samples of the form $(x_i, y_i)$;
**Parameters:** $\varepsilon, \eta_0, \eta_1, \eta_2, \eta_3 > 0.$;
1. Construct estimates $T_0, \ldots, T_{2\ell+2}$ for $\hat{f}_0, \hat{f}_1, \ldots, \hat{f}_{2\ell+2}$ using the first $\frac{N}{2}$ samples.
2. Let $S = \{(\tilde{w}_i, \tilde{a}_i, \tilde{b}_i)\}$ be the output of Algorithm 4 on inputs $T_0, \ldots, T_{2\ell+2}$ when run with parameters $\eta_0, \eta_1, \eta_2, \eta_3 > 0$.
3. For each $(x_i, y_i)$ and $i \in [N/2 + 1, N]$ construct the feature mapping
$\phi(x_i) = (Z(x_i), Z'(x_i))$ as described in the proof of Lemma D.12.
4. Set $\tau = 20m(8|S| + d)B\sqrt{\log(\frac{mdB|S|}{\varepsilon})}$ and find, via projected gradient descent, a vector $\hat{\beta}$ such that

$$\hat{L}_\tau(\hat{\beta}) \leq \min_{\beta:\|\beta\| \leq \sqrt{8|S|} + m(1+B)} \hat{L}_\tau(\beta) + \frac{\varepsilon^2}{100}.$$

Here $\hat{L}_\tau(\beta)$ is defined as

$$\hat{L}_\tau(\beta) = \frac{2}{N} \sum_{i=\frac{N}{2}+1}^{N} (y_i - \beta^\top \phi(x_i))^2 \mathbb{1}\left(\|\phi(x_i)\| < \tau\right).$$

**Result:** The function $g(x) = \hat{\beta}^\top \phi(x)$.

*Proof.* By triangle inequality,

$$\left|\sum_{i\in[n]} a_i\right| \leq \sum_{i\in[n]} |a_i| \leq n \max_{i\in[n]} |a_i| \leq n \sum_{i\in[n]} |a_i| \Rightarrow \left|\sum_{i\in[n]} a_i\right|^{2p} \leq n^{2p} \max_{i\in[n]} |a_i|^{2p} \leq n^{2p} \sum_{i\in[n]} |a_i|^{2p}$$

$\square$

Equipped with the essential claim, we are now ready to prove Lemma D.3.

**Lemma D.3.** *For any $\eta > 0$, if $T_k$ is estimated from $N \geq c_k d^k m^2 B^4 \text{poly}(\log(mdB/\eta))/\eta^2$ samples, then for some constant $c_k > 0$ that depends only on $k$, we have with probability at least $1 - (mdB)^{-\log(md)}$,*

$$\|T_k - \hat{f}_k\|_F \leq \eta. \tag{36}$$

*Proof.* Consider $p \in \mathbb{N}$, and a sum $S_Y = \sum_{j=1}^{N} Y_j$ of independent zero-mean r.v.s with $\frac{1}{N}\sum_{j=1}^{N} \mathbb{E}[Y_j^{2p}] \leq A_{2p}$ and $\frac{1}{N}\sum_{j=1}^{N} \mathbb{E}[Y_j^2] \leq A_2$. Then by Rosenthal's inequality (and Markov's inequality)

$$\mathbb{E}\left[\left(\sum_{j=1}^{N} Y_j\right)^{2p}\right] \leq 2^{p\log(p)+2p+p^2} \cdot \max\{NA_{2p}, (NA_2)^p\} \tag{37}$$

And, $\mathbb{P}\left[\left|\frac{1}{N}\sum_{j=1}^{N} Y_j\right| > \eta\right] \leq 2^{p\log(p)+2p+p^2} \cdot \max\left\{\frac{A_{2p}}{N^{2p-1}\eta^{2p}}, \left(\frac{A_2}{N\eta^2}\right)^p\right\}. \tag{38}$

Consider a fixed $\alpha \in [d]^k$ (an index of the tensor corresponding to the $k$th Hermite coefficient); $|\alpha| = k$. Given samples $\{(x^{(j)}, f(x^{(j)}) : j \in [N]\}$, the random variables of interest are $Z_j, Y_j$ are

$$Z_j = \sum_{i=1}^{m} a_i \sigma(w_i^\top x^{(j)} + b_i) He_\alpha(x^{(j)}), \quad \text{and} \quad Y_j := Z_j - \mathbb{E}[Z_j].$$

We will apply the above concentration inequality with the random variables $Y_j$. We need bounds for $\mathbb{E}[Y_j^2]$ and $\mathbb{E}[Y_j^{2p}]$. For convenience let $Z := \sum_{i=1}^{m} a_i \sigma(w_i^\top x + b_i) He_\alpha(x)$, and $Y := Z - \mathbb{E}[Z]$.

Note that by applying Claim D.2, we can bound these quantities as

$$\mathbb{E}[Y^{2p}] = \mathbb{E}[(Z - \mathbb{E}[Z])^{2p}] \leq 2^{2p}(\mathbb{E}[Z^{2p}] + \mathbb{E}[Z]^{2p}), \quad \text{where}$$

$$|\mathbb{E}[Z]| = \Big| \sum_{i=1}^{m} a_i He_{k-2}(b_i) \cdot \frac{\exp(-b_i^2/2)}{\sqrt{2\pi}} \cdot \prod_{t=1}^{k} w_i(\alpha(t)) \Big| \leq mB\sqrt{k!},$$

$$\mathbb{E}[Z^{2p}] = \mathbb{E}\Big[ \Big( \sum_{i=1}^{m} a_i \sigma(w_i^\top x + b_i) \Big)^{2p} He_\alpha(x)^{2p} \Big],$$

On the other hand, from Hölder's inequality, we have $(\sum_{i=1}^{m} |c_i| |z_i|)^{2p} \leq (\|c\|_{q^*}^{q^*})^{2p/q^*} \cdot \|z\|_{2p}^{2p}$ where $q^*$ is the dual norm of $2p$ i.e., $2p/q^* = 2p - 1$. Hence, again combined with Claim D.2, we have

$$\mathbb{E}[Z^{2p}] \leq \mathbb{E}\Big[ \Big( \sum_{i=1}^{m} a_i^{q^*} \Big)^{2p/q^*} \Big( \sum_{i=1}^{m} \sigma(w_i^\top x + b_i)^{2p} \Big) He_\alpha(x)^{2p} \Big] \leq (m^{2p-1}B^{2p}) \sum_{i=1}^{m} \mathbb{E}\Big[ (w_i^\top x + b_i)^{2p} He_\alpha(x)^{2p} \Big]$$

$$\leq (2^{2p}m^{2p-1}B^{2p}) \sum_{i=1}^{m} \mathbb{E}\Big[ ((w_i^\top x)^{2p} + b_i^{2p}) He_\alpha(x)^{2p} \Big]$$

$$\leq (2mB)^{2p} \sum_{i=1}^{m} \Big( \mathbb{E}\Big[ (w_i^\top x)^{2p} He_\alpha(x)^{2p} \Big] + B^{2p} \mathbb{E}\Big[ He_\alpha(x)^{2p} \Big] \Big)$$

We note that $w_i^\top x$ is a standard Gaussian since $\|w_i\|_2 = 1$.

Now, let $He_\alpha(x)$ involve different indices of $x$ up to $k_1, k_2, \ldots, k_d$ times. Note that $\sum_{t \in [d]} k_t \leq |\alpha| = k$. Using properties of Hermite polynomials, we can bound $\mathbb{E}[He_\alpha(x)^{2p}]$ as

$$\mathbb{E}[He_\alpha(x)^{2p}] = \mathbb{E}\Big[ \Big( \sum_{\sum_{t \in [d]} k_t \leq k} c_{k_1 \ldots k_d} \prod_{t \in [d]} x_t^{k_t} \Big)^{2p} \Big]$$

$$\leq \binom{d+k}{d}^{2p} \max_{\sum_{t \in [d]} k_t \leq k} c_{k_1 \ldots k_d}^{2p} \mathbb{E}\Big[ \prod_{t \in [d]} x_t^{2pk_t} \Big] \leq \binom{d+k}{d}^{2p} (2pk-1)!!(k!)^{2p}$$

$$\leq \Big( \binom{d+k}{d} \cdot k! \Big)^{2p} (2pk)^{pk} = C_1^{2p} \cdot (2pk)^{pk}$$

by setting $C_1 = \binom{d+k}{d} \cdot k!$ and repetitively applying Claim D.2. A similar argument also holds for $\mathbb{E}[(w_i^\top x)^{2p} He_\alpha(x)^{2p}]$ by Cauchy–Schwarz inequality

$$\mathbb{E}[(w_i^\top x)^{2p} He_\alpha(x)^{2p}] \leq \sqrt{\mathbb{E}[(w_i^\top x)^{4p}] \mathbb{E}[He_\alpha(x)^{4p}]} \leq \sqrt{\mathbb{E}[(w_i^\top x)^{4p}]} \binom{d+k}{d}^{2p} (k!)^{2p} (4pk)^{pk}$$

$$\leq \Big( \sqrt{4p} \cdot \binom{d+k}{d} \cdot k! \Big)^{2p} (4pk)^{pk} = (2^{k+2}p)^p \cdot C_1^{2p} \cdot (2pk)^{pk}$$

since $w_i^\top x$ follows a standard Gaussian distribution. Hence we have

$$\mathbb{E}[Z^{2p}] \leq C_1^{2p}(2mB)^{2p}(2pk)^{pk}((2^{k+2}p)^p + B^{2p})m$$

$$A_{2p} = \mathbb{E}[Y^{2p}] \leq 2^{2p}\Big( (2mBC_1)^{2p}(2pk)^{pk}((2^{k+2}p)^p + B^{2p})m + (mB\sqrt{k!})^{2p} \Big)$$

$$\implies A_{2p} \leq (C_2(pk)^{k/2}mB^2)^{2p}$$

where $C_2 = 8C_1^{2k}$. Note that $p = 1$ also gives the required bounds for $\mathbb{E}[Y^2]$.

Now, setting $p := \frac{1}{2}(\log(1/\eta) + \log(mdB))$, and applying Rosenthal's inequality (37) with $N = \frac{c'(k)}{\eta^2}m^2B^4\text{poly}(\log(mdB/\eta))$, we have for an appropriate constant $c'(k) > 0$

$$\mathbb{P}\left[\left|\frac{1}{N}\sum_{j=1}^{N}Y_j\right| > \eta\right] \leq 2^{p\log(p)+2p+p^2} \cdot \max\left\{\frac{A_{2p}}{N^{2p-1}\eta^{2p}}, \left(\frac{A_2}{N\eta^2}\right)^p\right\}$$

$$\leq 2^{p\log(p)+2p+p^2} \cdot \max\left\{\frac{(C_2mB^2)^{2p}(pk)^{pk}}{N^{2p-1}\eta^{2p}}, \left(\frac{(C_2mB^2k^{k/2})^2}{N\eta^2}\right)^p\right\}$$

$$\leq \left(\frac{1}{mdB}\right)^{\log(mdB)+\log(1/\eta)},$$

as required. Finally by setting $\eta = \eta'/\sqrt{d^k}$ and a union bound over all entries, we get that w.h.p., $\|T_k - \hat{f}_k\|_F \leq \eta'$, as required.

$\square$

## D.2 Recovering the parameters under errors

Suppose $\varepsilon > 0$ is the desired recovery error. The $m$ hidden units are split into groups

$$G = \left\{i \in [m] \mid |b_i| < c_\ell\sqrt{\log\left(\frac{1}{\varepsilon mdBs_m(W^{\odot\ell})}\right)}\right\}, \text{ and } P = \{1, 2, \ldots, m\} \setminus G. \tag{39}$$

where $c_\ell$ is an appropriate constant that depends only on the constant $\ell > 0$. Note that under the assumption that $s_m(W^{\odot\ell}) \geq 1/\text{poly}(m, d, B)$ in Theorem D.1, this reduces to

$$G = \left\{i \in [m] \mid |b_i| < c'_\ell\sqrt{\log(1/\varepsilon mdB)}\right\}.$$

We aim to recover all of the parameters of the units corresponding to $G$. For the terms in $P$, we will learn a linear function that approximates the total contribution from all the terms in $P$.

### D.2.1 Recovery of weight vectors $w_i$ for the terms in $G$

We first state the following important lemma showing that Jennrich's algorithm run with an appropriate choice of rank $k$ will recover each large term up to a sign ambiguity.

**Lemma D.4.** *Suppose $\varepsilon_2 \in (0, \frac{1}{4})$, and $\ell_1, \ell_2 \geq \ell, \ell_3 > 0$ be constants for some fixed $\ell$, and $T = \text{flatten}(\hat{f}_{\ell_1+\ell_2+\ell_3}, \ell_1, \ell_2, \ell_3)$ have decomposition $T = \sum_{i=1}^{m} \lambda_i(u_i \otimes v_i \otimes z_i)$ with $\lambda_i \in \mathbb{R}$ and unit vectors $u_i = w_i^{\otimes\ell_1} \in \mathbb{R}^{d^{\ell_1}}, v_i = w_i^{\otimes\ell_2} \in \mathbb{R}^{d^{\ell_2}}, z_i = w_i^{\otimes\ell_3} \in \mathbb{R}^{d^{\ell_3}}$. There exists $\eta_1 = \text{poly}(\varepsilon_2, s_m(W^{\odot\ell}))/\text{poly}(m, d, B) > 0$ and $\varepsilon_1 := \max\{2\varepsilon_2, 1/\text{poly}(1/\varepsilon_2, 1/s_m(W^{\odot\ell}), d^{\ell_1+\ell_2+\ell_3}, B)\}$ such that if*

$$\|T - \widetilde{T}\|_F \leq \eta'_1 := \min\left\{\text{poly}(\varepsilon_2, s_m(W^{\odot\ell}))/\text{poly}(m, d, B, 1/\eta_1), \frac{\eta_1}{2}\right\}, \tag{40}$$

*then Jennrich's algorithm runs with rank $k' := \text{argmax}_{r\leq m} s_r(\text{flatten}(\widetilde{T}, \ell_1, \ell_2 + \ell_3, 0)) > \eta_1$ and w.h.p. outputs[6] $\{\widetilde{\lambda}_i, \widetilde{w}_i\}_{i\in[k']}$ such that there exists a permutation $\pi : [m] \to [m]$ and signs $\xi_i \in \{1, -1\} \, \forall i \in [m]$ satisfying:*

$$(i) \ \forall i \in [m], \ |\lambda_i - \widetilde{\lambda}_{\pi(i)}| \leq \varepsilon_2^2, \text{ and} \tag{41}$$

$$(ii) \ \forall i \in [m], \text{ s.t. } |\lambda_i| > \varepsilon_1, \text{ we have } \|w_i^{\otimes t} - \xi_{\pi(i)}^t \widetilde{w}_{\pi(i)}^{\otimes t}\|_2 \leq \varepsilon_2, \quad \forall t \in [2\ell]. \tag{42}$$

Before we proceed to the proof of this lemma, we first state and prove a couple of simple claims. We use the following simple claim about the assumptions of the theorem implying lower bounds on the least singular value of the submatrices given by two columns of $W$.

---

[6]Note that one can also choose to pad the output with zeros to output $m$ sets of parameters instead of $k'$ if required.

**Claim D.5.** *Suppose the matrix $M_j \in \mathbb{R}^{d^j \times 2}$ formed by columns $u^{\otimes j}$ and $v^{\otimes j}$ for $u, v \in \mathbb{S}^{d-1}$. Suppose $s_2(M_\ell) \geq \kappa$, then $s_2(M_1) \geq \kappa/\sqrt{2\ell}$.*

*Proof.* Suppose $v = \alpha u + \sqrt{1 - \alpha^2} u^\perp$ for some $u^\perp \in \mathbb{S}^{d-1}$ that is perpendicular to $u$. It is easy to see that

$$\langle v^{\otimes \ell}, u^{\otimes \ell} \rangle = \alpha^\ell.$$

For two unit vectors $u, v$, the least singular value of the matrix given by them as columns is

$$\min_{\substack{x,y \in \mathbb{R} \\ x^2 + y^2 = 1}} \|xu + yv\|_2 = \min_{\substack{x,y \in \mathbb{R} \\ x^2 + y^2 = 1}} \sqrt{x^2 + y^2 + 2xy\langle u, v \rangle} = \min_{\substack{x,y \in \mathbb{R} \\ x^2 + y^2 = 1}} \sqrt{1 + 2xy\langle u, v \rangle} = \sqrt{1 - |\langle u, v \rangle|}.$$

$$\text{Hence, } \kappa^2 = 1 - \alpha^\ell \implies s_2([u\ v]) = \sqrt{1 - \alpha} = \sqrt{1 - (1 - \kappa^2)^{1/\ell}} \geq \frac{\kappa}{\sqrt{2\ell}}.$$

$\square$

We use the following simple claim shows that if we obtain a rank-1 term which is close, then the corresponding vectors are also close.

**Claim D.6.** *For any $\varepsilon > 0, \ell \in \mathbb{N}$ with $\ell \geq 2$, suppose $\alpha, \beta > 0$, and $u, v \in \mathbb{S}^{d-1}$ satisfy $\|\alpha u^{\otimes \ell} - \beta v^{\otimes \ell}\|_F \leq \varepsilon$, for some $\varepsilon \in [0, \alpha/2)$. Then there exists $\xi \in \{+1, -1\}$ such that for any $t \in \{1, 2, \ldots, \ell\}$, $\|u^{\otimes t} - \xi^t v^{\otimes t}\|_F \leq \sqrt{2}\varepsilon/\alpha$. Also $|\alpha - \beta| \leq 3\varepsilon$.*

We remark that if $\ell$ is odd, we can additionally conclude that $\xi = +1$, but this is not used in the arguments, so we skip its proof.

*Proof.* Suppose $A_1 = u^{\otimes t}, B_1 = v^{\otimes t}$ and $A_2 = u^{\otimes \ell - t}, B_2 = v^{\otimes \ell - t}$. Note that they all have unit norm. Let $\eta = \min\{\|A_1 - B_1\|_F, \|A_1 + B_1\|_F\}$. Then $A_1 = \sqrt{1 - \eta^2/2} B_1 + \frac{\eta}{\sqrt{2}} B_1^\perp$ for some $B_1^\perp$ with unit norm orthogonal to $B_1$. We have

$$\alpha u^{\otimes \ell} - \beta v^{\otimes \ell} = \alpha A_1 \otimes A_2 - \beta B_1 \otimes B_2 = \alpha \sqrt{1 - \frac{\eta^2}{2}} B_1 \otimes A_2 + \alpha \cdot \frac{\eta}{\sqrt{2}} B_1^\perp \otimes A_2 - \beta B_1 \otimes B_2$$

$$\text{Hence } \varepsilon^2 = \|\alpha u^{\otimes \ell} - \beta v^{\otimes \ell}\|_F^2 \geq \frac{\alpha^2 \eta^2}{2} \|B_1^\perp \otimes A_2\|_F^2 \geq \frac{\alpha^2 \eta^2}{2} \quad (\text{ since } B_1 \perp B_1^\perp).$$

Hence $\eta^2 \leq 2\varepsilon^2/\alpha^2$. For even $t$, it is easy to see that $\|u^{\otimes t} - v^{\otimes t}\|_F \leq \|u^{\otimes t} + v^{\otimes t}\|_F$; hence $\|u^{\otimes t} - v^{\otimes t}\|_F \leq \sqrt{2}\varepsilon/\alpha$. For odd $t$, it could be either $\|u^{\otimes t} - v^{\otimes t}\|_F = \eta$ or $\|u^{\otimes t} - v^{\otimes t}\|_F = \eta$; moreover the sign (in front of $v^{\otimes t}$) is coordinated across the different $t$ since $\|u^{\otimes t} - v^{\otimes t}\|_F^2 + \|u^{\otimes t} - v^{\otimes t}\|_F^2 = 2$. Hence for an appropriate sign $\xi \in \{+1, 1\}$ we have $\|u^{\otimes t} - \xi^t v^{\otimes t}\|_F = \eta$.

Finally, to give an upper bound on $|\alpha - \beta|$, we use the conclusion from the above bound with $t = 1$, to argue that for some $v^\perp \in \mathbb{S}^{d-1}$ that is orthogonal to $v$

$$\varepsilon^2 \geq \left\| \alpha \left( \sqrt{1 - \frac{\eta^2}{2}} \cdot v + \frac{\eta}{\sqrt{2}} v^\perp \right)^{\otimes \ell} - \beta v^{\otimes \ell} \right\|_F^2$$

$$= \left| \alpha \left( 1 - \frac{\eta^2}{2} \right)^{\ell/2} - \beta \right|^2 + \alpha^2 \left( 1 - \left( 1 - \frac{\eta^2}{2} \right)^\ell \right).$$

Since $\eta^2 \in (0, 1)$, we can use a simple linear approximation to claim that $t\eta^2/4 \leq |1 - (1 - \eta^2/2)^t| \leq t\eta^2$ for any $t > 0$. Hence, we get that

$$\varepsilon^2 \geq (|\alpha - \beta| - |\alpha|\ell\eta^2)^2 + \alpha^2 \left( \tfrac{1}{4}\ell\eta^2 \right).$$

Hence $|\alpha - \beta| \leq \varepsilon + |\alpha|\sqrt{\ell}\eta$, and $|\alpha|\sqrt{\ell}\eta \leq 2\varepsilon$.

Hence the claim follows.

$\square$

We now proceed to the proof of Lemma D.4.

*Proof of Lemma D.4.* The proof proceeds by first identifying a tensor $\widetilde{T}$ which we show satisfies all the conditions for Jennrich's robust algorithmic guarantee (Theorem A.5) with rank $k'$, which corresponds to the $k'$-th largest $|\lambda_i|$. Note that the recovery error in the rank-1 terms may be larger than some of the $k'$ terms of $\widetilde{T}$ (for example if there is not much separation between the $k'$ largest and $(k'+1)$th largest of the $\{|\lambda_i|\}$). Therefore, we argue that if $|\lambda_i|$ is sufficiently large, it will be recovered up to small error.

We start with some notation. Suppose $s_{\min} := s_m(W^{\odot \ell})$. Let $\widetilde{M} = \text{flatten}(\widetilde{T}, \ell_1, \ell_2 + \ell_3, 0)$ and $M = \text{flatten}(T, \ell_1, \ell_2 + \ell_3, 0)$. Set $U = W^{\odot \ell_1}, V = W^{\odot \ell_2}, Y = \text{diag}(\lambda)Z = \text{diag}(\lambda)W^{\odot \ell_3}$. Recall that $k' := \text{argmax}_{r \leq m} s_r(\widetilde{M}) > \eta_1$. Note that $M = U\text{diag}(\lambda)(V \odot Y)^\top$, where $\lambda = (\lambda_1, \ldots, \lambda_m)$. We remark that by Claim A.4

$$s_m(U) \geq \frac{s_m(W^{\odot \ell})}{(2m)^{\ell_1 - \ell}} \geq \frac{s_{\min}}{(2m)^{\ell_1}}, \quad \text{and similarly } s_m(V) \geq \frac{s_{\min}}{(2m)^{\ell_2}}, \quad s_m(V \odot Z) \geq \frac{s_{\min}}{(2m)^{\ell_2 + \ell_3}}.$$
(43)

We first argue that there are at least $k'$ values of $|\lambda_i|$ that are non-negligible. Since $\|T - \widetilde{T}\|_F \leq \eta_1' < \eta_1/2$, we have from Weyl's inequality that $s_{k'}(M) > \eta_1/2$. Let $S \subset [m]$ denote the indices corresponding to the largest $k'$ values of $|\lambda_i|$ (this is for analysis). The rank-1 terms restricted to $S$ will constitute the "ground-truth" decomposition $\widetilde{T}$. We first observe that

$$\min_{i \in S} |\lambda_i| > \frac{\eta_1}{2m}, \quad \text{and} \tag{44}$$

$$\forall i \in [m] \text{ s.t. } |\lambda_i| \geq \frac{\eta_1(2m)^{\ell_1 + \ell_2 + \ell_3}}{s_{\min}^2}, \quad \text{we have } i \in S. \tag{45}$$

To see why (44) holds, note that

$$\frac{\eta_1}{2} < s_{k'}(M) = s_{k'}\left(U\text{diag}(\lambda)(V \odot Y)^\top\right) \leq s_{k'}(\text{diag}(\lambda)) \cdot s_1(U) \cdot s_1(V \odot Y) \leq m \cdot \min_{i \in S} |\lambda_i|,$$

where we used the fact that all the columns of $U$ and $V \odot Z$ are unit vectors. To show (45), suppose we assume for contradiction that $|\lambda_i| > 2(2m)^{\ell_1 + \ell_2 + \ell_3}\eta_1/s_{\min}^2$, but $i \notin S$. Let $S' = S \cup \{i\}$. Then we can see that at least $k' + 1$ singular values of $\widetilde{M}$ are greater than $\eta_1$ since by Weyl's inequality,

$$s_{k'+1}(\widetilde{M}) \geq s_{k'+1}(M) - \eta_1' = s_{k'+1}\left(U\text{diag}(\lambda)(V \odot W)^\top\right)\eta_1'$$
$$\geq s_m(U) \cdot s_{k'+1}(\text{diag}(\lambda)) \cdot s_m(V \odot W) \geq \frac{s_{\min}}{(2m)^{\ell_1}} \cdot |\lambda_i| \cdot \frac{s_{\min}}{(2m)^{\ell_2}}$$
$$\geq 2\eta_1 - \eta_1' > \eta_1.$$

Hence (44) and (45) are both true.

We now argue that we satisfy the requirements of Theorem A.5 (the robust guarantee). Let $U_S, V_S$ and $Y_S$ denote the restriction of the factor matrices $U, V, Y$ to the columns corresponding to $S$. Then by Claim A.4

$$s_{k'}(U) \geq s_m(U) \geq \frac{s_m(W^{\odot \ell})}{(2m)^{\ell_1 - \ell}} \geq \frac{s_{\min}}{(2m)^{\ell_1}}, \text{ and } s_{k'}(U) \geq s_m(U) \geq \frac{s_m(W^{\odot \ell})}{(2m)^{\ell_2 - \ell}} \geq \frac{s_{\min}}{(2m)^{\ell_2}}.$$

Moreover for any two columns $i, j \in S$, we have that the restriction of $Z$ to these two columns $Y_{\{i,j\}}$ satisfies

$$s_2(Y_{\{i,j\}}) \geq \min\{|\lambda_i|, |\lambda_j|\} \cdot s_2(W_{\{i,j\}}) \geq \frac{\eta_1 \cdot s_{\min}}{\sqrt{2\ell} \cdot m}.$$

Moreover the maximum singular values of the factor matrices $U, V$ are all upper bounded by $\sqrt{m}$.

Finally, suppose $T_S = \sum_{i \in S} \lambda_i u_i \otimes v_i \otimes z_i$, then the error between the input tensor and $T_S$

$$\|\widetilde{T} - T_S\|_F \le \|\widetilde{T} - T\|_F + \|T - T_S\|_F \le \eta_1' + \Big\|\sum_{i \notin S} \lambda_i u_i \otimes v_i \otimes z_i\Big\|_F$$

$$= \eta_1' + \Big\|\mathrm{flatten}\Big(\sum_{i \notin S} \lambda_i u_i \otimes v_i \otimes z_i, \ell_1, \ell_2 + \ell_3, 0\Big)\Big\|_F$$

$$\le \eta_1' + \sqrt{m}s_1\Big(\mathrm{flatten}\Big(\sum_{i \notin S} \lambda_i u_i \otimes v_i \otimes z_i, \ell_1, \ell_2 + \ell_3, 0\Big)\Big)$$

$$\le \eta_1' + \sqrt{m}\eta_1 \le (\sqrt{m} + 1)\eta_1.$$

Now applying Theorem A.5 with $\varepsilon_{A.5} = \varepsilon_2^2$, and setting $\eta_1'$ such that $\eta_1' = \min\Big\{\eta_{A.5}\big(\varepsilon_{A.5} = \varepsilon_2^2, \kappa = \frac{\sqrt{m}(2m)^{\ell_1+\ell_2}}{s_{\min}}, d^{\ell_1+\ell_2+\ell_3}, \delta = \frac{\eta_1 \cdot s_{\min}}{\sqrt{2\ell} \cdot m}\big)/(\sqrt{m} + 1), \eta_1/2\Big\}$, we have that the rank-1 terms can be recovered up to accuracy $\varepsilon_2^2$ (up to renaming the indices $i \in [m]$):

$$\forall i \in S, \quad \|\lambda_i u_i \otimes v_i \otimes z_i - \widetilde{\lambda}_i \widetilde{u}_i \otimes \widetilde{v}_i \otimes \widetilde{z}_i\|_F \le \varepsilon_2^2. \tag{46}$$

$$\text{From Claim D.6}, \forall i \in S, \quad \|w_i - \xi_i \widetilde{w}_i\|_2 \le \frac{\varepsilon_2^2}{|\lambda_i|}, \tag{47}$$

for appropriate signs $\xi_i \in \{+1, -1\}$. We remark that the choice of $\eta_1'$ is consistent with both Theorem A.5 and this lemma, since in our case $\kappa \le \mathrm{poly}_\ell(m)/s_{\min}$ and $1/\delta \le \mathrm{poly}_\ell(m, 1/\eta_1)/s_{\min}$. If $s_{\min} > 0$ becomes too small, we will directly set $\eta_1'$ as $\eta_1/2$ instead.

Note that from (46) and triangle inequality, we already have for $i \in S$

$$|\lambda_i - \widetilde{\lambda}_i| = \Big|\|\lambda_i u_i \otimes v_i \otimes z_i\|_F - \|\widetilde{\lambda}_i \widetilde{u}_i \otimes \widetilde{v}_i \otimes \widetilde{z}_i\|_F\Big| \le \|\lambda_i u_i \otimes v_i \otimes z_i - \widetilde{\lambda}_i \widetilde{u}_i \otimes \widetilde{v}_i \otimes \widetilde{z}_i\|_F \le \varepsilon_2^2.$$

(For terms not in $S$, the output $\widetilde{\lambda}_i = 0$ and $|\lambda_i| < \eta_1/(2m)$, hence it is still satisfied). For any $i \in [m]$ s.t., $|\lambda_i| > \varepsilon_1$, we have from (45) that $i \in S$; hence

$$\text{For all } i \text{ s.t. } |\lambda_i| > \varepsilon_1, \quad \|w_i - \xi_i \widetilde{w}_i\|_2 \le \frac{\varepsilon_2^2}{\varepsilon_1} \le \varepsilon_2,$$

as long as $|\varepsilon_1| \ge \sqrt{\varepsilon_2}$. A similar proof also holds for $\|w_i^{\otimes t} - \xi_i^t \widetilde{w}_i^{\otimes t}\|_F$ by using Claim A.4 with general $t \ge 1$ in (47). This completes the proof.

$\square$

A direct application of Lemma D.4 establishes the following claim, showing that we can recover all the weight vectors $w_i$ for each term $i \in G$ up to a sign ambiguity.

**Lemma D.7.** *For any $\varepsilon_2 > 0$, there exists an $\eta_2' = \frac{\mathrm{poly}(\varepsilon_2, s_m(W^{\odot \ell}))}{\mathrm{poly}_\ell(m, d, B)} > 0$ such that if the estimates $\|T_k - \hat{f}_k\|_F \le \eta_2'$ for all $k \in \{0, 1, \ldots, 2\ell + 2\}$, then steps 1-4 of Algorithm 4 finds a set $\{\widetilde{w}_i : i \in [m']\}$ such that there exists a one-to-one map $\pi : [m'] \to [m]$ satisfying (i) every $i \in G$ has a pre-image in $\pi$ (i.e., every term in $G$ is recovered), and for appropriate signs $\{\xi_i \in \{1, -1\} : i \in [m']\}$,*

$$\forall i \in [m'], \forall t \in [2\ell], \quad \|\xi_i^t \widetilde{w}_i^{\otimes t} - w_{\pi(i)}^{\otimes t}\|_F \le \varepsilon_2. \tag{48}$$

*In particular $\forall i \in [m']$, we have $\|\xi_i \widetilde{w}_i - w_{\pi(i)}\|_2 \le \varepsilon_2$.*

The stronger guarantee for all $t \in [2\ell]$ will be useful in bounding the recovery error of the $a_i, b_i$ in later steps.

*Proof of Lemma D.7.* The proof uses the robust guarantees of Jennrich's algorithm in Lemma D.4 along with the crucial property of separation of roots in Lemma 4.3.

Set $\varepsilon_1' := \varepsilon_1/\sqrt{k!/2}$ and $\varepsilon_1 = \mathrm{poly}(\varepsilon_2, m, d, s_m(W^{\odot \ell}))$ be given by Lemma D.4. Similarly $\eta_2'$ is specified by the requirement of Lemma D.4. Set also $\eta_2 = 4\varepsilon_1$.

Consider a $\widetilde{w}_i$ output by the algorithm in step 4; and suppose w.l.o.g. it was output in step 2. Then we have that $|\widetilde{\lambda}_i| \geq \eta_2$. Further, $|\widetilde{\lambda}_i - \lambda_i| \leq \varepsilon_2^2 < \eta_2/4$. Hence, for every term $i \in [m']$ that is output after step 4, we have $|\lambda_i| > \eta_2/2 \geq \varepsilon_1$.

We first argue that every term in $G$ is one of the $m'$ terms output by the algorithm in step 4. Consider the decompositions of the two tensors obtained from the Hermite coefficients of $f$ i.e.,

$$\hat{f}_{2\ell+1} = \sum_{i=1}^{m} (-a_i) \cdot He_{2\ell-1}(b_i) \cdot \frac{1}{\sqrt{2\pi}} \exp(-b_i^2/2) \cdot w_i^{\otimes 2\ell+1} = \sum_{i=1}^{m} \lambda_i (w_i^{\otimes \ell}) \otimes (w_i^{\otimes \ell}) \otimes w_i \tag{49}$$

$$\hat{f}_{2\ell+2} = \sum_{i=1}^{m} a_i \cdot He_{2\ell}(b_i) \cdot \frac{1}{\sqrt{2\pi}} \exp(-b_i^2/2) \cdot w_i^{\otimes 2\ell+2} = \sum_{i=1}^{m} \lambda_i' (w_i^{\otimes \ell}) \otimes (w_i^{\otimes \ell}) \otimes w_i^{\otimes 2}. \tag{50}$$

Note that from Lemma 4.3 we have that for every $x \in \mathbb{R}$, at least one of $|He_{2\ell+2}(x)|, |He_{2\ell+1}(x)|$ is at least $\sqrt{k!/2}$. Moveover since $i \in G$ for our choice of $c_k$ in (39), we have that $e^{-b_i^2/2}/\sqrt{2\pi} > \varepsilon_1'$. Hence for each $i \in G$, we have that $\max\{|\lambda_i|, |\lambda_i'|\} > \varepsilon_1$.

Finally, we now prove that when $|\lambda_i| \geq \varepsilon_1$, the corresponding $w_i$ is recovered up to error $\varepsilon_2$. From Lemma D.4, if $\widetilde{w}_i$ is the vector output by one of the decompositions for $w_i$, we have for all $t \in [2\ell]$ that $\|w_i^{\otimes t} - \xi_i^t \widetilde{w}_i^{\otimes t}\|_F \leq \varepsilon_2$ for some sign $\xi_i \in \{1, -1\}$ as required. Moreover since $\eta_2 := \varepsilon_1/2$ and the error in each rank-1 term is at most $\varepsilon_{A.5} < \eta_1/2$, we have that none of these terms are removed as duplicates of other terms. On the other hand, since $\eta_3 := 2\varepsilon_2$, we have that duplicates are correctly removed. Hence we have that for every $i \in [m']$, $\|w_i^{\otimes t} - \xi_i^t \widetilde{w}_i^{\otimes t}\|_F \leq \varepsilon$ for appropriate signs $\xi_i \in \{\pm 1\}$.

$\square$

### D.2.2  Recovering error for the parameters $a_i, b_i$ for terms $i \in G$.

The following lemmas now proves the recovery for each $i \in G$, the $a_i$ (no sign ambiguity) and the $b_i$ up to the same sign ambiguity as in $w_i$ (and in fact, this holds for all the terms output in steps 1-5 of Algorithm 4).

Before we start the proof of the main lemmas, we first show a key property of Hermite polynomials we will utilize later.

**Claim D.8.** $\forall x \in \mathbb{R}, |He_k(x)| \exp(-x^2/2) \leq \sqrt{k!}$

*Proof.* We utilize Cramér's inequality for Hermite functions that for all $x \in \mathbb{R}$, $|\psi_k(x)| \leq \pi^{-1/4}$, where $\psi_k(x)$ is the $k$'th Hermite function given by

$$|\psi_k(x)| = (2^k k! \sqrt{\pi})^{-1/2} \exp(-x^2/2) |H_k(x)|$$

with $H_k(x)$ denoting the $k$'th physicist's Hermite polynomial[7]. Now, substituting $x$ with $x/\sqrt{2}$ yields

$$|\psi_k(\frac{x}{\sqrt{2}})| = (k! \sqrt{\pi})^{-1/2} \exp(-x^2/4) |He_k(x)| \leq \pi^{-1/4}$$

$$\implies \exp(-x^2/2) |He_k(x)| \leq \sqrt{k!} \exp(-x^2/4) \leq \sqrt{k!}$$

$\square$

With this claim, we are now ready to proceed.

**Lemma D.9.** *For $\varepsilon > 0$ in the definition of $G$ in (39), there exists $\eta_3' = \frac{\text{poly}(\varepsilon, s_m(W^{\otimes \ell}))}{\text{poly}_\ell(m, d, B)} > 0$, and $\varepsilon_3' = \frac{\text{poly}(\varepsilon, s_m(W^{\otimes \ell}))}{\text{poly}_\ell(m, d, B)} > 0$ such that for some $\xi_i \in \{\pm 1\} \, \forall i \in [m']$*

$$\text{if } \|T_k - \hat{f}_k\|_F \leq \eta_3' \, \forall k \leq 2\ell+2, \quad \text{and} \quad \|\widetilde{w}_i^{\otimes t} - \xi^t w_i^{\otimes t}\|_F \leq \varepsilon_3', \forall i \in [m'], \forall \ell \leq t \leq \ell+3.$$

*then steps 5-6 of the algorithm finds $(\widetilde{a}_i, \widetilde{b}_i : i \in [m'])$ such that*

$$|\widetilde{a}_i - a_i| \leq \varepsilon, \text{ and } |\widetilde{b}_i - \xi_i b_i| \leq \varepsilon. \tag{51}$$

---

[7]The physicist's Hermite polynomials are defined as $H_k(x) = \frac{(-1)^k}{\exp(-x^2)} \cdot \frac{d^k}{dx^k} \exp(-x^2)$

Note that in Lemma D.7 we showed that $G$ is contained in the $m'$ terms output in steps 1-4 (and hence step 5 as well).

The above uses the following two lemmas which gives a robust version of Lemma 4.5 when there are errors in the estimates. We remark that $\beta = \alpha e^{-z^2/2}$ in the notation of Lemma 4.5.

**Lemma D.10** (Robust version of Lemma 4.5 for $k \geq 2$). *Suppose $k \in \mathbb{N}, k \geq 2, B \geq 1$, and $\alpha, z \in \mathbb{R}$ be unknown parameters. There exists a constant $c_k = c(k) \geq 1$ such that for any $\varepsilon \in (0, \frac{1}{4})$ satisfying (i) $|\alpha| \in [\frac{1}{B}, B]$ and $|z| \leq B$, and (ii) $|z| < 2\sqrt{\log(c_k/\varepsilon^{1/4}(1+B)^3))}$, if we are given values $\gamma_k, \gamma_{k+1}, \gamma_{k+2}, \gamma_{k+3}$ s.t. for some $\xi \in \{\pm 1\}$,*

$$\left|\gamma_j - \alpha \cdot \frac{(-1)^j}{\sqrt{2\pi}} e^{-z^2/2} He_j(\xi z)\right| \leq \varepsilon' = \frac{\varepsilon^4}{2(1+B)^2} \quad \forall j \in \{k, k+1, k+2, k+3\},$$

*then the estimates $\widetilde{z}, \widetilde{\alpha}$ obtained as:*

$$\widetilde{z} = -\frac{\gamma_{q+1} + q \cdot \gamma_{q-1}}{\gamma_q} \text{ where } q := \underset{j \in \{k+1, k+2\}}{\operatorname{argmax}} |\gamma_j|, \text{ and } \widetilde{\alpha} = (-1)^q \frac{\sqrt{2\pi}\gamma_q}{e^{-\frac{\widetilde{z}^2}{2}} He_q(\widetilde{z})}$$

$$\text{satisfy } |\widetilde{z} - \xi z| \leq \frac{\varepsilon|z|}{B+1} \leq \varepsilon, \text{ and } |\widetilde{\alpha} - \alpha| \leq \varepsilon. \tag{52}$$

*Proof.* Set $\varepsilon' := \varepsilon^4/(2(1+B)^2)$, and let $\beta := \alpha e^{-z^2/2}/\sqrt{2\pi}$. Under our assumptions $|\beta| > c_k(\varepsilon')^{1/4}(1+B)^2$. We use the following fact about Hermite polynomials:

$$He_{r+1}(z) = zHe_r(z) - r \cdot He_{r-1}(z). \tag{53}$$

For convenience, for a scalar quantity $v$ we denote by $v = a \pm \varepsilon$ iff $|v - a| \leq \varepsilon$. Recall that $q = \operatorname{argmax}_{j \in \{k+1, k+2\}} |\gamma_j|$. Since $\beta\sqrt{k!/2} > 4\varepsilon'$ by the conditions, we have that $|\gamma_q| > \beta\sqrt{k!}/2$.

Setting $r = q$ in (53) and dividing by $He_q(z)$ we get using its odd or even function property depending on parity of $q$,

$$\xi z = \xi \cdot \frac{He_{q+1}(z) + qHe_{q-1}(z)}{He_q(z)} = \frac{\beta He_{q+1}(\xi z) + q \cdot \beta He_{q-1}(\xi z)}{\beta He_q(\xi z)}$$

$$= -\frac{(\gamma_{q+1} \pm \varepsilon') + q(\gamma_{q-1} \pm \varepsilon')}{\gamma_q \pm \varepsilon'} = -\frac{\gamma_{q+1} + q\gamma_{q-1}}{\gamma_q(1 \pm \frac{\varepsilon'}{\gamma_q})} \pm \frac{(k+3)\varepsilon'}{\gamma_q(1 \pm \frac{\varepsilon'}{\gamma_q})}$$

$$= -\frac{\gamma_{q+1} + q \cdot \gamma_{q-1}}{\gamma_q} \cdot \left(1 \pm \frac{2\varepsilon'}{|\gamma_q|}\right) \pm \frac{(k+3)\varepsilon'}{\gamma_q(1 \pm \frac{\varepsilon'}{|\gamma_q|})}$$

$$= \widetilde{z}(1 \pm \sqrt{\varepsilon'}) \pm \sqrt{\varepsilon'}, \quad \text{since } \beta\sqrt{k!}/2 \geq 2(k+3)\sqrt{\varepsilon'}.$$

Let $g(z) = e^{-z^2/2} He_q(z)/\sqrt{2\pi}$. Hermite polynomials satisfy $He_q'(x) = qHe_{q-1}(x)$. Hence

$$g'(z) = e^{-z^2/2}\big(qHe_{q-1}(z) - zHe_q(z)\big)/\sqrt{2\pi} = -\frac{e^{-z^2/2} He_{q+1}(z)}{\sqrt{2\pi}},$$

by applying (53). Also, by Claim D.8, $\max_{z'} |g'(z')| \leq \max_{z'} e^{-z'^2/2}|He_{q+1}(z')|/\sqrt{2\pi} \leq (q+1)!$. Hence,

$$|g(\widetilde{z}) - g(\xi z)| \leq \max_{z' \in [z, \widetilde{z}] \cup z' \in [\widetilde{z}, z]} |g'(z)||\widetilde{z} - \xi z| \leq 4(q+1)!(1+B)\sqrt{\varepsilon'}$$

Plugging these error bounds into $\alpha$, and using $He_q(\xi z) = \xi^q He_q(x)$ we have

$$\widetilde{\alpha} = \frac{\gamma_q}{g(\widetilde{z})} = \frac{\alpha \xi^q \cdot \frac{e^{-z^2/2}}{\sqrt{2\pi}} He_q(z) \pm \varepsilon'}{\frac{e^{-z^2/2} He_q(\xi z)}{\sqrt{2\pi}} \pm |g(\widetilde{z}) - g(\xi z)|}$$

$$= \frac{\alpha g(z) \pm \varepsilon'}{g(z) \pm 4(q+1)!(1+B)\sqrt{\varepsilon'}} = \alpha\Big(1 \pm \frac{8(q+1)!(1+B)\sqrt{\varepsilon'}}{|g(z)|}\Big) \pm \frac{2\varepsilon'}{|g(z)|}$$

$$|g(z)| = \frac{|\beta He_q(z)|}{|\alpha|} \geq \frac{|\beta|\sqrt{k!}}{2B} > c_k(1+B)^2(\varepsilon')^{1/4}$$

Hence $|\widetilde{\alpha} - \alpha| \leq 8|\alpha| \times \dfrac{(q+1)!(1+B)\sqrt{\varepsilon'}}{c_k(1+B)^2(\varepsilon')^{1/4}} + \dfrac{2\varepsilon'}{c_k(1+B)^2\varepsilon'^{1/4}} \leq (\varepsilon')^{1/4} \leq \varepsilon,$

because of our choice of $c_k = 16(k+3)!$.

$\square$

The simpler variant of the above lemma (Lemma D.10) for $k = 1$ which is used in the full-rank setting, follows a very similar analysis and is stated below.

**Lemma D.11** (Robust version of Lemma 4.5 for $k = 1$). *Suppose $B \geq 1$, and $\alpha, z \in \mathbb{R}$ be unknowns. There exists a constant $c \geq 1$ such that for any $\varepsilon \in (0, \frac{1}{4})$ satisfying (i) $|\alpha| \in [\frac{1}{B}, B]$ and $|z| \leq B$, and (ii) $|z| < 2\sqrt{\log(c/\varepsilon^{1/4}(1+B)^3))}$, if we are given values $\gamma_0, \gamma_1$ s.t. for some $\xi \in \{\pm 1\}$,*

$$\Big|\gamma_j - \alpha \cdot \frac{(-1)^j}{\sqrt{2\pi}} e^{-z^2/2} He_j(\xi z)\Big| \leq \varepsilon' = \frac{\varepsilon^4}{2(1+B)^2} \quad \forall j \in \{0, 1\},$$

*then the estimates $\widetilde{z}, \widetilde{\alpha}$ obtained as:*

$$\widetilde{z} = -\frac{\gamma_1}{\gamma_0}, \quad \text{and} \quad \widetilde{\alpha} = \frac{\sqrt{2\pi}\gamma_0}{e^{-\frac{\widetilde{z}^2}{2}}}$$

*satisfy* $\ \Big|\widetilde{z} - \xi z\Big| \leq \dfrac{\varepsilon|z|}{B+1} \leq \varepsilon, \quad \text{and} \quad \big|\widetilde{\alpha} - \alpha\big| \leq \varepsilon.$ (54)

Note that $He_0(z) = 1$ and $He_1(z) = z$ to see the similarities between Lemma D.10 and Lemma D.11

*Proof.* Set $\varepsilon' := \varepsilon^4/(2(1+B)^2)$, and let $\beta := \alpha e^{-z^2/2}/\sqrt{2\pi}$. Under our assumptions $|\beta| > c(\varepsilon')^{1/4}(1+B)^2$. For convenience we denote for a scalar $v$, $v = a \pm \varepsilon$ iff $|v - a| \leq \varepsilon$.

Since $\beta > 4\varepsilon'$ by the conditions, we have that $|\gamma_0| > \beta$. Recall that $He_0(z) = 1$ and $He_1(z) = z$. Hence,

$$\xi z = \xi \cdot \frac{He_1(z)}{He_0(z)} = \frac{\beta He_1(\xi z)}{\beta He_0(z)}$$

$$= -\frac{(\gamma_1 \pm \varepsilon')}{\gamma_0 \pm \varepsilon'} = -\frac{\gamma_1 \pm \varepsilon'}{\gamma_0(1 \pm \frac{\varepsilon'}{\gamma_0})} = -\frac{\gamma_1}{\gamma_0} \cdot \Big(1 \pm \frac{2\varepsilon'}{|\gamma_0|}\Big) \pm \frac{\varepsilon'}{\gamma_0(1 \pm \frac{\varepsilon'}{|\gamma_0|})}$$

$$= \widetilde{z}(1 \pm \sqrt{\varepsilon'}) \pm \sqrt{\varepsilon'}, \quad \text{since } \beta \geq 8\sqrt{\varepsilon'}.$$

To argue about $|\widetilde{\alpha} - \alpha|$, let $g(z) = e^{-z^2/2}/\sqrt{2\pi}$. Its derivative $g'(z)$ satisfies by Claim D.8, $\max_{z'} |g'(z')| \leq \max_{z'} |z'|e^{-z'^2/2}/\sqrt{2\pi} \leq 1$. Hence,

$$|g(\widetilde{z}) - g(\xi z)| \leq \max_{z' \in [z, \widetilde{z}] \cup z' \in [\widetilde{z}, z]} |g'(z)||\widetilde{z} - \xi z| \leq 4(1+B)\sqrt{\varepsilon'}$$

Plugging these error bounds into $\alpha$ we have

$$\widetilde{\alpha} = \frac{\gamma_0}{g(\widetilde{z})} = \frac{\alpha \cdot \frac{e^{-z^2/2}}{\sqrt{2\pi}} \pm \varepsilon'}{\frac{e^{-z^2/2}}{\sqrt{2\pi}} \pm |g(\widetilde{z}) - g(\xi z)|}$$

$$= \frac{\alpha g(z) \pm \varepsilon'}{g(z) \pm 4(1+B)\sqrt{\varepsilon'}} = \alpha\left(1 \pm \frac{8(1+B)\sqrt{\varepsilon'}}{g(z)}\right) \pm \frac{2\varepsilon'}{g(z)}$$

$$g(z) = \frac{\beta}{\alpha} \geq \frac{|\beta|}{B} > c(1+B)^2(\varepsilon')^{1/4}$$

$$\text{Hence } |\widetilde{\alpha} - \alpha| \leq 8|\alpha| \times \frac{(1+B)\sqrt{\varepsilon'}}{c(1+B)^2(\varepsilon')^{1/4}} + \frac{2\varepsilon'}{c(1+B)^2\varepsilon'^{1/4}} \leq (\varepsilon')^{1/4} \leq \varepsilon,$$

because of our choice of $c = 16$.

$\square$

We now prove Lemma D.9.

*Proof of Lemma D.9.* Set $\varepsilon_3 = \varepsilon^4 s_m(W^{\otimes \ell}/(16m^{3/2}(1+B)^2)$. For each of the $j \in \{\ell, \ell+1, \ell+2, \ell+2\}$, we have from Lemma 3.5 that

$$\hat{f}_j = \sum_{i=1}^{m} (-1)^j \cdot a_i \cdot He_{j-2}(b_i) \cdot \frac{\exp(\frac{-b_i^2}{2})}{\sqrt{2\pi}} \cdot w_i^{\otimes j}.$$

Moreover $\|T_j - \hat{f}_j\|_F \leq \eta_3'$. Also for the terms $i \notin G$, we have for each $\ell \leq j \leq \ell+3$, the corresponding term

$$\left| a_i \cdot \frac{\exp(-b_i^2/2)}{2\pi} \cdot He_{j-2}(b_i) \right| \leq B \cdot (\varepsilon m d B)^{c_\ell^2/2} < \frac{\varepsilon_3}{2m}.$$

$$\text{Hence } \left\| T_j - \sum_{i \in G} (-1)^j \cdot a_i \cdot He_{j-2}(b_i) \cdot \frac{\exp(\frac{-b_i^2}{2})}{\sqrt{2\pi}} \cdot w_i^{\otimes j} \right\|_F \leq \varepsilon_3' + m \cdot \frac{\varepsilon_3}{2m} \leq \varepsilon_3,$$

where the first line follows from our choice of $\varepsilon_3$ and our choice of $c_\ell$.

Let $m' := |G|$. Next we establish that the linear system is well-conditioned for each $\ell \leq j \leq \ell+3$. For any signs $\xi_i \in \{\pm 1\} \; \forall i \in [m]$, the matrix formed by the vectors $\{\xi_i^\ell w_i^{\otimes \ell} : i \in G\}$ has non-negligible least singular value. Moreover from Claim A.4 (applied three times), we have for $\ell \leq j \leq \ell+3$, we have the matrix formed by columns $\{\xi_i^j w_i^{\otimes j} : i \in G\}$ has least singular value $s_{m'}\big((w_i^{\otimes j} : i \in G)\big) \geq s_m(W^{\otimes \ell})/(2m)^{3/2}$. Suppose $M_j, \widetilde{M}_j \in \mathbb{R}^{d^j \times m'}$ be the matrices with the $i$th columns $(\xi_i w_i)^{\otimes j}$ and $\widetilde{w}_i^{\otimes j}$ respectively for $i \in G$. Then by Weyl's inequality, we have

$$s_{m'}(\widetilde{M}_j) \geq s_{m'}(M_j) - \|\widetilde{M}_j - M_j\|_F \geq \frac{s_m(W^{\otimes \ell})}{(2m)^{3/2}} - \sum_{i \in G} \|\widetilde{w}_i^{\otimes j} - \xi_i w_i^{\otimes j}\|_F$$

$$\geq \frac{s_m(W^{\otimes \ell})}{(2m)^{3/2}} - m \cdot \varepsilon_3' \geq \frac{s_m(W^{\otimes \ell})}{4m^{3/2}}, \quad \text{since } \varepsilon_3 < \frac{s_m(W^{\otimes \ell})}{4m^2}.$$

The target solution to the linear system for each $\ell \leq j \leq \ell+3$

$$\forall i \in G, \; \zeta_j^*(i) := a_i \cdot (-1)^j \xi_i^j \cdot \frac{\exp(\frac{-b_i^2}{2})}{\sqrt{2\pi}} He_{j-2}(b_i).$$

Note that since for any $j$, $\sup_{z \in \mathbb{R}} e^{-z^2/2} He_j(z) \leq c'_j$ for some bounded constant $c'_j < \infty$. Now a standard error analysis of the linear system yields (see e.g., [Bha97]) we have for all $\ell \leq j \leq \ell + 3$,

$$\|\zeta_j - \zeta_j^*\|_2 \leq \left(s_{m'}(\widetilde{M}_j)\right)^{-1} \left(\varepsilon_3 + \|M_j - \widetilde{M}_j\|_F \|\zeta_j^*\|_2\right)$$

$$\leq \frac{4m^{3/2}}{s_m(W^{\otimes \ell})} \left(\varepsilon_3 + c'_j \cdot B \cdot \sqrt{m'} \varepsilon'_3\right).$$

Hence, $\forall i \in G, \ \left|\zeta_j(i) - \zeta_j^*(i)\right| \leq \frac{8m^{3/2}}{s_m(W^{\otimes \ell})} \cdot \varepsilon_3 \leq \frac{\varepsilon^4}{2(1+B)^2}.$

since $\varepsilon'_3 \leq \frac{1}{2}(c'_{\ell+3} B \sqrt{m}) \varepsilon_3$, and for our choice of $\varepsilon$.

Finally we can now apply Lemma D.10 for $\ell \geq 2$ or Lemma D.11 for $\ell = 1$ for each of the $i \in G$ separately with $\gamma_j = \zeta_j(i)$ (note that the error is at most $\varepsilon'$ as in Lemmas D.10 and D.11). The output is $\widetilde{a}_i = \alpha, \widetilde{b}_i = z$ and conclude that $|\widetilde{a}_i - a_i| \leq \varepsilon$, and $|\widetilde{b}_i - \xi_i b_i| \leq \varepsilon$.

$\square$

### D.3 Learning Guarantees via Linear Regression

In the previous sections we designed algorithms based on tensor decompositions that, given i.i.d. samples from a network $f(x) = \sum_{i=1}^m a_i \sigma(w_i^\top x + b_i)$, can recover approximations (up to signs) for "good units", i.e, $G = \{i \in [m] : |b_i| < O(\sqrt{\log(\frac{1}{\varepsilon m d B})})\}$. In this section we will show how to use these approximations to perform improper learning of the target network $f(x)$ via a simple linear regression subroutine. Our algorithm will output a functions of the form $g(x) = \sum_{i=1}^{m'} a'_i \sigma(w_i'^\top x + b'_i) + w''^\top x + C$, where $m' \leq 8m$. In particular we will prove the following.

**Lemma D.12.** *Let $\varepsilon > 0$ and $f(x) = \sum_{i=1}^m a_i \sigma(w_i^\top x + b_i)$ be an unknown target network. Let $S$ be a given set of tuples of the form $(\widetilde{w}_i, \widetilde{b}_i, \widetilde{a}_i)$ with $\|\widetilde{w}_i\| = 1$, such that for each $i \in G$, there exists $j \in S$, and $\xi_{j_1}, \xi_{j_2}, \xi_{j_3} \in \{-1, +1\}$, such that $\|w_i - \xi_{j_1} \widetilde{w}_j\| \leq O(\frac{\varepsilon}{mdB})$, $|b_i - \xi_{j_2} \widetilde{b}_j| \leq O(\frac{\varepsilon}{mdB})$, and $|a_i - \xi_{j_3} \widetilde{a}_j| \leq O(\frac{\varepsilon}{mdB})$. Then for any $\delta \in (0,1)$, given $N = \text{poly}(m, d, B, \frac{1}{\varepsilon}, \log(\frac{1}{\delta}))$ i.i.d. samples of the form $(x, y = f(x))$ where $x \sim N(0, I)$, there exists an algorithm (Algorithm 5) that runs time polynomial in $N$ and with probability at least $1 - \delta$ outputs a network $g(x)$ of the form $g(x) = \sum_{i=1}^{m'} a'_i \sigma(w_i'^\top x + b'_i) + w''^\top x + C$, where $m' \leq 8|S|$, such that*

$$\mathbb{E}_{x \sim \mathcal{N}(0, I_{d \times d})} \left(f(x) - g(x)\right)^2 \leq \varepsilon^2.$$

*Furthermore, when $\xi_{j_1} = \xi_{j_2}$ and $\xi_{j_3} = +1$ for all $j \in S$ (i.e., the sign ambiguity of $w_i$ and $b_i$ are the same, and there is no ambiguity in the sign of $a_i$ for all $i \in G$), then the number of hidden units in $g(x)$ is at most $|S| + 2$.*

While the above lemma is more general, when it is applied in the context of Theorem 3.2 it satisfies the conditions of the "furthermore" portion of the lemma. Our algorithm for recovering $g(x)$ will set up a linear regression instance in an appropriate feature space. In order to do this we will need the lemma stated below that shows that there is a good linear approximation for the units not in $G$, i.e., $P = [m] \setminus G$.

**Lemma D.13** (Approximating $f_P$). *Let $c > 2$ be a fixed constant. Consider $f_P(x) = \sum_{i=1}^m a_i \sigma(w_i^\top x + b_i) \mathbb{1}\left(|b_i| \geq c \sqrt{\log(\frac{1}{\varepsilon m d B})}\right)$. Then there exists a function $g_P(x) = \beta_P^\top x + C_P$ where $\|\beta_P\| \leq mB$ and $|C_P| \leq mB^2$ such that for a constant $c' > 0$ that depends on $c$,*

$$\mathbb{E}_{x \sim \mathcal{N}(0, I)}[f_P(x) - g_P(x)]^2 = c' \varepsilon^2.$$

Before proceeding to the proof of the main lemmas, we first establish an auxiliary claim that we will utilize. The following claim shows how one can combine some of the activation units output by the regression step to get a ReLU network with at most $|G| + 2$ units.

**Claim D.14.** *Given a function $g(x)$ of the form*

$$g(x) = v^\top x + c + \sum_{i \in m'} \alpha_i \sigma(w_i^\top x + b_i) + \alpha'_i \sigma(-w_i^\top x - b_i), \tag{55}$$

*then $g(x)$ can be expressed as a ReLU network with at most $m' + 2$ activation units as*

$$g(x) = \beta_0 \sigma(w_0^\top x + b_0) - \beta_0 \sigma(-w_0^\top x - b_0) + \sum_{i=1}^{m'} \beta_i \sigma(w_i^\top x + b_i), \tag{56}$$

*where for each $i \in [m']$, $\beta_i = \alpha_i + \alpha_i'$ and $w_0 \in \mathbb{S}^{d-1}, b_0 \in \mathbb{R}, \beta_0 \in \mathbb{R}$ chosen to satisfy $\beta_0 w_0 = v - \sum_{i=1}^{m'} \alpha_i' w_i$ and $\beta b_0 = c - \sum_{i-1}^{m'} \alpha_i' b_i$.*

*Proof.* First we note that for any $z \in \mathbb{R}$, $\sigma(z) = \frac{1}{2}(|z| + z)$ and $\sigma(-z) = \frac{1}{2}(|z| - z)$. Hence we have

$$\gamma \sigma(z) + \gamma' \sigma(-z) = \tfrac{1}{2}(\gamma + \gamma')|z| + \tfrac{1}{2}(\gamma - \gamma')z, \text{ and } z = \sigma(z) - \sigma(-z). \tag{57}$$

Hence the terms are consolidated by replacing terms of the form $\sigma(w_i^\top x + b_i)$ and $\sigma(-w_i^\top x - b_i)$ by one ReLU unit so that the coefficient of $|w_i^\top x + b_i|$ match, along with a linear term. All the linear terms are themselves consolidated together, and replaced by a sum of two ReLU units. Now, substituing the setting of $\beta_i$ $w_0, b_0$ in (56) and simplifying, we have

$$\beta_0 \sigma(w_0^\top x + b_0) - \beta_0 \sigma(-w_0^\top x - b_0) + \sum_{i=1}^{m'} \beta_i \sigma(w_i^\top x + b_i)$$

$$= \beta_0 (w_0^\top x + b_0) + \sum_{i=1}^{m'} (\alpha_i + \alpha_i') \cdot \frac{1}{2} \left( |w_i^\top x + b_i| + (w_i^\top x + b_i) \right)$$

$$= v^\top x + c + \sum_{i=1}^{m'} \frac{1}{2} (\alpha_i + \alpha_i')|w_i^\top x + b_i| + \frac{1}{2}(\alpha_i - \alpha_i')(w_i^\top x + b_i)$$

$$= v^\top x + c + \sum_{i=1}^{m'} \alpha_i \sigma(w_i^\top x + b_i) + \alpha_i' \sigma(-w_i^\top x - b_i) = g(x),$$

where the last line follows from (57) and (55). $\qquad \square$

We are now ready to prove the main lemmas. We first establish the main result assuming the lemma above and provide a proof of the lemma at the end of the subsection.

*Proof of Lemma D.12.* In order to find the approximate network $g(x)$ we will set up a linear regression problem in an appropriate feature space. We begin by describing the construction of the feature space and showing that there does indeed exist a linear function in the space that approximates $f(x)$. We first focus on the terms in the set $G$, i.e.,

$$f_G(x) = \sum_{i \in G} a_i \sigma(w_i^\top x + b_i).$$

In order to approximate $f_G(x)$ we create for each $(\widetilde{w}_j, \widetilde{b}_j, \widetilde{a}_j) \in S$, eight features $Z_{j,1}, \ldots, Z_{j,8}$ where each feature is of the form $\xi_{j_3} \widetilde{a}_j \sigma(\xi_{j_1} \widetilde{w}_j^\top x + \xi_{j_2} \widetilde{b}_j)$ for $\xi_{j_1}, \xi_{j_2}, \xi_{j_3} \in \{-1, +1\}$. Consider a particular $i \in G$. Since the set $S$ consists of a good approximation $(\widetilde{w}_j, \widetilde{b}_j, \widetilde{a}_j)$ for the unit $i$, it is easy to see that one of the eight features corresponding to $Z_{j,:}$ approximates the $i$th unit well (by matching the signs appropriately). In other words we have that there exists $r \in [8]$ such that

$$|Z_{j,r} - a_i \sigma(w_i^\top x + b_i)| = |a_j' \sigma(w_j'^\top x + b'_j) - a_i \sigma(w_i^\top \cdot x + b_i)| \tag{58}$$

$$\leq |(a_j' - a_i)\sigma(w_i^\top x + b_i)| + |a_j'(\sigma(w'^\top_j x + b'_j) - \sigma(w_i^\top x + b_i))| \tag{59}$$

$$\leq O\left(\frac{\varepsilon}{mdB}\right)(B + \|x\|) + O\left(\frac{\varepsilon}{mdB}\right) + O\left(\frac{\varepsilon}{mdB}\right)\|x\|. \tag{60}$$

Noting that $\mathbb{E}[\|x\|^2] = d$ we get that there exists a vector $\beta_1^*$ with $\|\beta_1^*\|_2 \leq \sqrt{8|S|}$ in the feature space $Z(x)$ defined as above such that

$$\mathbb{E}\left[f_1(x) - \beta_1^{*\top} Z(x)\right]^2 \leq \frac{\varepsilon^2}{100}. \tag{61}$$

To approximate terms not in $G$, i.e., $f_P(x) = \sum_{i \notin G} a_i \sigma(w_i^\top x + b_i)$, we use Lemma D.13 to get that there exists a vector $\beta_2^*$ with $\|\beta_2^*\| \leq m(1+B)$ in the $Z' = (x, 1)$ feature space such that

$$\mathbb{E}\big[f_P(x) - \beta_2^{*\top} Z'(x)\big]^2 \leq \frac{\varepsilon^2}{100}. \tag{62}$$

Combining the above and noting that $y = f(x) = f_G(x) + f_P(x)$, we get that there exists a vector $\beta^*$ in the $\phi(x) = (Z(x), Z'(x))$ feature space with $\|\beta^*\| \leq \sqrt{8|S|} + m(1+B)$ such that

$$\mathbb{E}\big[y - \beta^{*\top} \phi(x)\big]^2 \leq \frac{\varepsilon^2}{20}. \tag{63}$$

In order to approximate $\beta^*$ we solve a truncated least squares problem. In particular, define the truncated squared loss $L_\tau(\beta) = \mathbb{E}[(y - \beta^\top \phi(x))^2 \mathbb{1}(\|\phi(x)\| < \tau)]$. Furthermore we define the empirical counter part $\hat{L}_\tau(\beta)$ based on $N$ i.i.d. samples drawn from the distribution of $\phi(x)$. For an appropriate value of $\tau$ we will output $\hat{\beta}$ such that

$$\hat{L}_\tau(\hat{\beta}) \leq \min_{\beta : \|\beta\| \leq \sqrt{8|S|} + m(1+B)} \hat{L}_\tau(\beta) + \frac{\varepsilon^2}{100}. \tag{64}$$

In particular we will set $\tau = 20m(8|S|+d)B\sqrt{\log(\frac{mdB|S|}{\varepsilon})}$. Notice that the empirical truncated loss above is convex and for the chosen value of $\tau$, has gradients bounded in norm by $\text{poly}(m, d, B, |S|, \frac{1}{\varepsilon})$. Hence we can use the projected gradient descent algorithm [BBV04] to obtain a $\hat{\beta}$ that achieves the above guarantee in $N \cdot \text{poly}(m, d, B, |S|, \frac{1}{\varepsilon})$ time. Furthermore using standard uniform convergence bounds for bounded loss functions [MRT18] we get that if $N = \text{poly}(m, d, B, |S|, \frac{1}{\varepsilon}, \log(\frac{1}{\delta}))$ then with probability at least $1 - \delta$ we have

$$L_\tau(\hat{\beta}) \leq \min_{\beta : \|\beta\| \leq \sqrt{8|S|} + m(1+B)} L_\tau(\beta) + \frac{\varepsilon^2}{50} \tag{65}$$

$$\leq L_\tau(\beta) + \frac{\varepsilon^2}{50}. \tag{66}$$

Finally, it remains to relate the truncated loss $L_\tau(\beta)$ to the true loss $L(\beta) = \mathbb{E}[y - \beta^\top \phi(x)]^2$. We have that for any $\beta$ such that $\|\beta\|_2 \leq \sqrt{8|S|} + m(1+B)$,

$$|L_\tau(\beta) - L(\beta)| = \mathbb{E}[(y - \beta^\top \phi(x))^2 \mathbb{1}(\|\phi(x)\| \geq \tau)]. \tag{67}$$

Next notice that if $\|\phi(x)\| \geq 2^j \tau$ then we must have that either $|\tilde{a}_j \sigma(\tilde{w}_j^\top \cdot x + \tilde{b}_j)| \geq \frac{2^j \tau}{8|S|+d}$ or that for some $i \in [d]$, $|x_i| \geq \frac{2^j \tau}{8|S|+d}$. For our choice of $\tau$, this probability is bounded by $(8|S| + d)e^{-2^{2j} \Omega(\log(\frac{mdB|S|}{\varepsilon}))}$. Hence we get that

$$|L_\tau(\beta) - L(\beta)| = \mathbb{E}[(y - \beta^\top \phi(x))^2 \mathbb{1}(\|\phi(x) \geq \tau\|)] \tag{68}$$

$$= \sum_{j=0}^{\infty} \mathbb{E}[(y - \beta^\top \phi(x))^2 \mathbb{1}(\|\phi(x)\| \in [2^j \tau, 2^{j+1} \tau))] \tag{69}$$

$$\leq \sum_{j=0}^{\infty} O(2^{2j} m^2 (8|S| + d)^2 \tau^2)(8|S| + d)e^{-2^{2j} \Omega(\log(\frac{mdB|S|}{\varepsilon}))} \tag{70}$$

$$\leq \frac{\varepsilon^2}{50}. \tag{71}$$

Hence, the output network $g(x) = \hat{\beta} \cdot \phi(x) = \sum_{i=1}^{m'} a_i' \sigma(w_i'^\top x + b_i') + w''^\top x + C$ satisfies with probability at least $1 - \delta$ that

$$\mathbb{E}_{x \sim \mathcal{N}(0, I_{d \times d})}\big(f(x) - g(x)\big)^2 \leq \varepsilon^2.$$

Notice that since a linear function can be simulated via two ReLU units (see Claim D.14), our output function $g(x)$ is indeed a depth-2 neural network with $m' + 2 \leq 8m$ hidden units.

Furthermore, while the statement of Lemma D.12 assumes that the signs of units in $G$ are completely unknown, the output of the tensor decomposition procedure from Theorem 3.2 in fact recovers, for each $i \in G$, the signs of $a_i$ exactly and the signs of the corresponding $(w_i, b_i)$ are either both correct or both incorrect. Hence when applying Lemma D.12 to our application we only need to create two features for each unit in $G$. In other words we can output a network of the form $g(x) = w''^{\top} x + C + \sum_{i=1}^{m'} a_i' \sigma(w_i'^{\top} x + b_i') + a_i'' \sigma(-w_i'^{\top} x - b_i')$, where $m' \leq m$. Finally, from Claim D.14, the above network can be written as a depth-2 network with ReLU activations and at most $m + 2$ hidden units. $\qquad\square$

We end the subsection with the proof of Lemma D.13.

*Proof of Lemma D.13.* Consider a particular unit $i$ such that $b_i > c\sqrt{\log(\frac{1}{\varepsilon m d B})}$. Then notice that $z_i = w_i^{\top} x + b_i \sim \mathcal{N}(b_i, 1)$. By using standard properties of the Gaussian cdf, we get that by approximating $\sigma(z_i)$ by the linear term $z_i$ we incur the error

$$\mathbb{E}_{z_i \sim \mathcal{N}(b_i, 1)}(z_i - \sigma(z_i))^2 = \frac{1}{\sqrt{2\pi}} \int_{-\infty}^{0} z_i^2 e^{-(z_i - b_i)^2/2} dz_i \tag{72}$$

$$\leq O(b_i^2) e^{-b_i^2/2} \leq O\left(\frac{\varepsilon^2}{m^2}\right). \tag{73}$$

Similarly for a unit with $b_i < -c\sqrt{\log(\frac{1}{\varepsilon m d B})}$, by approximating $\sigma(z_i)$ with the constant zero function we incur the error

$$\mathbb{E}_{z_i \sim \mathcal{N}(b_i, 1)}(\sigma(z_i))^2 = \frac{1}{\sqrt{2\pi}} \int_{0}^{\infty} z_i^2 e^{-(z_i - b_i)^2/2} dz_i \tag{74}$$

$$\leq O(b_i^2) e^{-b_i^2/2} \leq O\left(\frac{\varepsilon^2}{m^2}\right). \tag{75}$$

Hence, each unit $i \in P$ with $z_i = w_i^{\top} x + b_i$ has a good linear approximation $\widetilde{z}_i = \widetilde{w}_i^{\top} x + \widetilde{b}_i$ of low error. Combining the above we get that

$$\mathbb{E}_{x \sim \mathcal{N}(0, I_{d \times d})}\left(\sum_{i \in P} a_i(z_i - \widetilde{z}_i)\right)^2 \leq c' \varepsilon^2, \tag{76}$$

for a constant $c'$ that depends on $c$. Furthermore it is easy to see that the linear approximation $\sum_{i \in P} a_i \widetilde{z}_i$ is of the form $\beta_P^{\top} x + C_P$ where $\|\beta_P\| \leq \sum_{i \in P} |a_i| \|w_i\| \leq mB$ and $|C_P| \leq \sum_{i \in P} |a_i b_i| \leq mB^2$. $\qquad\square$

## D.4 Wrapping up the proofs

With the lemmas above, we can now complete the proof of Theorem 3.1, Theorem D.1 and Theorem 3.2.

**Proof of Theorem D.1** We first set the parameters according to the polynomial bounds from the different lemmas in this section.

For the final error $\varepsilon$ in approximating $f$, we will set $\varepsilon' := \varepsilon/(4mB)$. Also set $\varepsilon_3', \eta_3'$ according to Lemma D.9 with the $\varepsilon$ in Lemma D.9 set to $\varepsilon'$. Then set $\varepsilon_2 = \varepsilon_3'$, and also set $\varepsilon_1 = \sqrt{\varepsilon_2}$. Now we can set the algorithm parameters $\eta_3 := 2\varepsilon_2$, and $\eta_2 := 4\varepsilon_1$, and $\eta_0 = \min\{\eta_3', \eta_2'\}$, where $\eta_2'$ is given by Lemma D.7. Moreover $\eta_1$ (and $\eta_1'$) are set according to Lemma D.4.

First by using Lemma D.3 we see that with $\text{poly}_{\ell}(d, m, B, 1/s_m(W^{\odot\ell}), 1/\varepsilon)$ we can estimate all the Hermite coefficients up to $2\ell + 2$ up to $\eta_0$ error in Frobenius norm. Then, for our setting of parameters we have from Lemma D.7 that for every $\widetilde{w}_i$ for $i \in [m']$ output by steps 1-4 of Algorithm 4, we have that there exists a $w_i$ (up to relabeling $i$) such that $\|w_i^{\otimes t} - \widetilde{w}_i^{\otimes t}\|_F \leq \varepsilon_2 = \varepsilon_3' < \varepsilon'$. Then we can apply Lemma D.9 to conclude that for all such terms $i \in [m']$ that are output we get estimates $\widetilde{a}_i, \widetilde{b}_i$ with $|\widetilde{a}_i - a_i| + |\widetilde{b}_i - b_i| \leq \varepsilon'$. Moreover using Lemma D.7 and Lemma D.9 also show that every $i \in G$ is also one of the $m'$ terms that are output. Hence for each $i \in \widetilde{G}$, we have recovered each parameter up to error $\varepsilon'$. This completes the proof.

**Proof of the full-rank setting: Theorem 3.1** The guarantees for Theorem 3.1 hold for the following Algorithm 6, which is a robust variant of Algorithm 1 in the special case of $\ell = 1$. It first uses Algorithm 4 to approximately recover for each $i \in [m]$, the $a_i$, and up to an ambiguity in a sign (captured by unknown $\xi_i \in \{1, -1\}$) close estimates of $w_i$ and $b_i$. Then it runs Algorithm 3 to disambiguate the sign by recovering $\xi_i$.

---

**Algorithm 6:** Robust full-rank algorithm: recover $\{a_i, b_i, w_i\}$ given estimates $\{T_0, T_1, \ldots, T_4\}$.

---

**Input:** Estimates $T_0, T_1, T_2, T_3, T_4$;
**Parameters:** $\eta_0, \eta_1, \eta_2, \eta_3 > 0$.;
1. Run Algorithm 4 on parameters $(\eta_0, \eta_1, \eta_2, \eta_3)$ with inputs $T_0, T_1, T_2, T_3, T_4$ to receive results $(\widetilde{w}_i, \widetilde{a}_i, \widetilde{b}_i)_{i \in [m]}$. Note that $\widetilde{b}_i, \widetilde{w}_i$ are only recovered up to signs.
2. Run Algorithm 3 (FixSigns) on parameters $m, T_1$ and $(\widetilde{a}_i, \widetilde{b}_i, \widetilde{w}_i : i \in [m])$ to recover $(\widetilde{a}_i, \widetilde{b}_i, \widetilde{w}_i : i \in [m])$.
**Result:** Output $\{\widetilde{w}_i, \widetilde{a}_i, \widetilde{b}_i : 1 \leq i \leq m\}$.

---

*Proof of Theorem 3.1.* We first set the parameters of Algorithm 6 as dictated by Theorem D.1 (and its proof) in the special case of $\ell = 1$. Let $\varepsilon_0 > 0$ be chosen so that

$$\varepsilon_0 < \frac{\min\{\Phi(-c\sqrt{\log(1/\varepsilon m d B)}), \varepsilon\} \cdot s_m(W)}{8\sqrt{m}B^2},$$

and $\eta_0$ to be the smaller of $\varepsilon_0/((1 + B)\sqrt{m})$, and whatever is specified Theorem D.1 for $\varepsilon = \varepsilon_0$. Note that $\Phi(-c\sqrt{\log(1/(\varepsilon m d B))}) \geq \Omega((\varepsilon m d B)^{c^2/2} \min\{1, \varepsilon m d B\})$.

We draw $N = \text{poly}(d, m, B, 1/s_m(W), 1/\varepsilon_0)$ i.i.d. samples and run Algorithm 4 with the parameters $\eta_1, \eta_2, \eta_3$ as described in the proof of Theorem D.1. From the assumptions of Theorem 3.1, we have that each $i \in [m]$ belongs to the "good set" $G$ as well. Hence, from the guarantee of Theorem D.1 we will obtain w.h.p. for each $i \in [m]$ estimates $\widetilde{a}_i, \widetilde{b}_i, \widetilde{w}_i$ (up to relabeling the indices $[m]$) such that up an unknown sign $\xi_i \in \{1, -1\}$ we have

$$|a_i - \widetilde{a}_i| + |\widetilde{b}_i - \xi_i b_i| + \|\widetilde{w}_i - \xi_i w_i\|_2 \leq \varepsilon_0 < \varepsilon. \tag{77}$$

Now consider the (ideal) linear system in the unknowns $\{z_i : i \in [m]\}$ given by $\widehat{f}_1 = \sum_{i=1}^m z_i(a_i \xi_i w_i)$; it has $d$ equations in $m \leq d$ unknowns. Let $M := W \text{diag}((\xi_i a_i : i \in [m]))$ be a $d \times m$ matrix representing the above linear system as $Mz = \widehat{f}_1$. From Lemma 3.5, $z_i^* = \xi_i \Phi(b_i)$ is a solution. Moreover $M$ is well-conditioned: since $|a_i| \in [1/B, B]$, we have $s_1(M) \leq B s_1(W) \leq B\sqrt{m}$, while $s_m(M) \geq s_m(W)/B$ (from the assumption on $W$). Hence, this is a well-conditioned linear system with a unique solution $z^*$.

Algorithm 3 solves the linear system $\widetilde{M}z = T_1$, where $\widetilde{M} = \widetilde{W} \text{diag}(\widetilde{a})$; here each column of $\widetilde{M}$ is close to its corresponding column of $M$, while the sample estimate $T_1$ for $\widehat{f}_1$ satisfies $\|T_1 - \widehat{f}_1\|_2 \leq \eta_0$. Let $\widetilde{z}$ be a solution to the system $\widetilde{M}z = T_1$.

Observe that if $\|z - \widetilde{z}\|_\infty \leq \|\widetilde{z} - z\|_2$ is at most $\min_i |z_i^*|$, then Algorithm 3 recovers the signs correctly, since $\widetilde{z}$ will not flip in sign. To calculate this perturbation first observe that $i$th column of $E = \widetilde{M} - M$ has length at most

$$\|\widetilde{a}_i \xi_i \widetilde{w}_i - a_i w_i\|_2 \leq |\widetilde{a}_i - a_i|\|\xi_i \widetilde{w}_i\|_2 + a_i\|\xi_i \widetilde{w}_i - w_i\|_2 \leq \varepsilon_0 + B\varepsilon_0 \leq \varepsilon_0(1 + B).$$

Hence $\|E\|_2 = s_1(E) \leq \varepsilon_0(1 + B)\sqrt{m}$. Moreover by Weyl's inequality $s_m(\widetilde{M}) \geq s_m(M) - \|E\| \geq \frac{1}{B}s_m(W) - \varepsilon_0(1 + B) \geq s_m(W)/(2B)$ due to our choice of parameter $\varepsilon_0$. From standard perturbation bounds for linear systems, we have

$$\|\widetilde{z} - z^*\|_2 \leq \left(s_m(\widetilde{M})\right)^{-1} \left(\|T_1 - \widehat{f}_1\|_2 + s_1(M - \widetilde{M})\|z^*\|_2\right)$$

$$\leq \frac{2B}{s_m(W)}\left(\eta_0 + \varepsilon_0(1 + B)\sqrt{m}\right) \leq \frac{4\varepsilon_0 B^2 m}{s_m(W)}$$

$$\leq \Phi(-c\sqrt{\log(1/(\varepsilon m d B))}) \leq \frac{1}{2}\min_{i \in [m]} |z_i^*|$$

as required, due to our choice of $\varepsilon_0$. Hence the signs are also recovered accurately. This along with (77) concludes the proof.

$\square$

**Proof of Theorem 3.2** In order to establish Theorem 3.2 we draw $N = \text{poly}_\ell(d, m, B, 1/s_m(W^{\odot \ell}), 1/\varepsilon)$ i.i.d. samples and run Algorithm 5 with the parameters $\eta_0, \eta_1, \eta_2, \eta_3$ as described in the proof of Theorem D.1. From the guarantee of Theorem D.1 we will obtain w.h.p., up to signs, approximations for all units in $G$ up to an error of $O(\frac{\varepsilon}{mdB})$. Furthermore, given these approximations the guarantee of Lemma D.12 tells us that w.h.p. the function $g(x)$ output by Algorithm 5 will satisfy $\mathbb{E}_{x \sim \mathcal{N}(0, I_{d \times d})} \big( f(x) - g(x) \big)^2 \leq \varepsilon^2$.

# E  Smoothed Analysis

We use the smoothed analysis framework of Spielman and Teng [ST04], which is a beyond-worst-case-analysis paradigm that has been used to explain the practical success of various algorithms. In smoothed analysis, the performance of the algorithm is measured on a small random perturbation of the input instance. We use the model studied in the context of parameter estimation and tensor decomposition problems to obtain polynomial time guarantees under non-degeneracy conditions [BCMV14, Vij20]. The smoothed analysis model for the depth-2 neural RELU network setting is as follows:

1. An adversary chooses set of parameters $a, b \in \mathbb{R}^m$ and $W \in \mathbb{R}^{d \times m}$.

2. The weight matrix $\widehat{W} \in \mathbb{R}^{d \times m}$ is obtained by a small *random* i.i.d. perturbation as $\widehat{W}_{ij} = W_{ij} + \xi_{i,j} \ \forall i \in [d], j \in [m]$ where $\xi_{i,j} \sim N(0, \tau^2/d)$. (Note that the average squared perturbation in each column is $\tau^2$) [8].

3. Each sample $(x, f(x))$ is drawn i.i.d. with $x \sim N(0, I_{d \times d})$ and $f(x) = a^\top \sigma(\widehat{W}^\top x + b)$.

The goal is to design an algorithm that with high probability, estimates the parameters $a, b, \widehat{W}$ up to some desired accuracy $\varepsilon$ in time $\text{poly}(m, d, 1/\varepsilon, 1/\tau)$. We now prove the following corollary of Theorem 3.2.

**Corollary 3.3** *Suppose $\ell \in \mathbb{N}$ and $\varepsilon > 0$ are constants in the smoothed analysis model with smoothing parameter $\tau > 0$, and also assume the ReLU network $f(x) = a^\top \sigma(\widehat{W}^\top x + b)$ is B-bounded with $m \leq 0.99 \binom{d+\ell-1}{\ell}$. Then there is an algorithm that given $N \geq \text{poly}_\ell(m, d, 1/\varepsilon, B, 1/\tau)$ samples runs in $\text{poly}(N, m, d)$ time and with high probability finds a ReLU network $g(x) = a'^\top \sigma(W'^\top x + b')$ with at most $m + 2$ hidden units such that the $L_2$ error $\mathbb{E}_{x \sim \mathcal{N}(0, I_{d \times d})}[(f(x) - g(x))^2] \leq \varepsilon^2$. Furthermore there are constants $c, c' > 0$ and signs $\xi_i \in \{\pm 1\} \ \forall i \in [m]$, such that in $\text{poly}(N, m, d)$ time, for all $i \in [m]$ with $|b_i| < c\sqrt{\log(1/(\varepsilon \cdot mdB))}$, we can recover $(\widetilde{a}_i, \widetilde{w}_i, \widetilde{b}_i)$, such that $|a_i - \widetilde{a}_i| + \|w_i - \xi_i \widetilde{w}_i\|_2 + |b_i - \xi_i \widetilde{b}_i| < c'\varepsilon/(mB)$.*

*Proof.* The proof of the corollary follows by combining Theorem 3.2 with existing results on smoothed analysis [BCPV19] on the least singular value $s_m(\widehat{W}^{\odot \ell})$. We apply Theorem 2.1 of [BCPV19] with $\rho = \tau$, $U$ being the identity matrix to derive that for any $\delta > 0$ and $m \leq (1 - \delta)\binom{d+\ell-1}{\ell}$, we get with probability at least $1 - m \exp(-\Omega_\ell(\delta n))$ that

$$s_m(\widehat{W}^{\odot \ell}) \geq \frac{c_\ell}{\sqrt{m}} \left(\frac{\tau}{d}\right)^\ell.$$

We then just apply Theorem 3.2 to conclude the proof.

$\square$

---

[8]Think of $\tau$ as a fairly small but inverse polynomial quantity $1/\text{poly}(n, d)$.