# OpenReview forum: "Efficient Algorithms for Learning Depth-2 Neural Networks with General ReLU Activations"
_NeurIPS.cc/2021/Conference — NeurIPS 2021 Poster_

### Official Review · Reviewer_yvHw · 2021-07-09

**Rating:** 7
**Confidence:** 4

**Summary:**

This paper gave a polynomial time and sample efficient algorithm for learning a two layer ReLU neural network with bias terms under standard Gaussian distribution. The result does not requires the weights $w_i$'s to be linearly independent. Instead, for any constant $\ell,$ as long as $w_i^{\otimes \ell}$'s are linearly independent, the algorithm can learn the network with time complexity and sample complexity exponential in $\ell$ and polynomial in other parameters. The algorithm relies on decomposing multiple higher order tensors ($O(l)$-order) arising from the Hermite expansion of the ground truth label function. It's also proved that the network is identifiable as long as no two weight vectors are parallel.

**Limitations And Societal Impact:**

The authors have adequately addressed the limitations and potential negative societal impact of their work.

**Main Review:**

I think this is a good theory paper that designed a tensor decomposition algorithm for learning two layer ReLU nets with bias terms. Most of the previous works along this line do not consider bias terms. The bias terms do bring some new challenges into the analysis. I believe the ideas and techniques in this paper is useful in the theoretical study of ReLU nets with bias terms.

The algorithm proceeds by first estimating several high order tensors of the weight vectors (with order $O(\ell)$), and then decompose these tensors to recover the weight vectors. With the recovered weight vectors, the second layer coefficients and bias terms can also be recovered. The bias terms in the ReLU indeed brings some challenges. First, the algorithms needs to recover the weights from multiple higher order tensors instead of a single one as in the bias-free setting. Second, when a bias term is high, it's impossible to estimate the corresponding weight vectors using polynomial number of samples. So the authors divided the weights into two groups, one group with relatively small bias and the other group with large bias; then recovered the weights in the first group, and fitted the second group using a linear function.

My main concern for this paper is that the analysis is restricted into the Gaussian inputs. It does not seem to be able to generalize to other distributions because it strongly relies on the Hermite polynomials. Also, these tensor decomposition algorithms are usually not very effective in real network training.

That being said, I still think this is a solid theory paper that studied an important problem. The techniques developed in this paper can be useful in the general study of ReLU nets with bias terms.

**Time Spent Reviewing:**

4

---

> ### Author Response · Authors · 2021-08-10
> **Author response to reviewer yvHw**
>
> We thank the reviewer for their feedback and comments.
>
> On the gaussianity assumption:
>
> We note that from a theoretical perspective even learning of ReLU networks under the Gaussian distribution is a challenging problem and most recent works on the topic make the same assumptions. Going beyond Gaussians is a natural direction for future work.
> One potential approach is to consider appropriate orthogonal polynomials as in [JSA15] instead of Hermite polynomials, and learn a network from those coefficients.

---

> > ### Comment · Reviewer_yvHw · 2021-08-28
> > **Thanks for the response**
> >
> > I have read the authors' response and other reviews. I think this is a solid theory paper on neural network optimization and I keep my positive review. Thanks!

---

### Official Review · Reviewer_pfsw · 2021-07-10

**Rating:** 7
**Confidence:** 3

**Summary:**

The paper gives a polynomial-time algorithm for learning depth-2 neural networks with the ReLU activation and bias terms in the realizable setting w.r.t. the square loss. The result assumes Gaussian marginals and certain assumptions on the weights matrix W. Assumptions of this nature are required due to known harness results. Prior works on efficient learning of depth-2 networks either assume that there is no bias or require some assumptions on the activation function that are not satisfied by the ReLU function. Thus, the algorithm shown in this paper is the first efficient algorithms for learning depth-2 ReLU networks with bias.

The main results are:
- Assume that the input distribution is Gaussian, the parameters of the target network are bounded, and the smallest singular value of W^{\odot \ell} for some constant \ell is at least inverse polynomial, then the algorithm runs in polynomial time in all relevant parameters (and in exponential time in \ell) and finds w.h.p. a depth-2 network that approximates the target network. If the bias terms of the target network are not too large, then the algorithm also recovers the parameters of the target network. If the bias terms are large then recovering (some of) the parameters is not possible. Where \ell=1 the assumption on W^{\odot \ell} implies that W has full rank. Where \ell>1 the requirement may allow the number of hidden neurons m to be significantly larger than the input dimension d.
- The authors show that the above assumption on W^{\odot \ell} holds w.h.p. in the smoothed-analysis setting, namely, where the target weights matrix is slightly perturbed with a random i.i.d. gaussian noise. Hence, in the smoothed-analysis setting the assumption on W^{\odot \ell} is not required. In this case, the number of hidden neurons can be m=O(d^\ell) and the running time of the algorithm is exponential in \ell (and polynomial in the other parameters).
- Another consequence of the analysis, is that if the weights in the second layer are non-zero, and there are no two parallel weight vectors in the fist layer, then the parameters of the target network are uniquely identifiable (not necessarily in polynomial time).

The algorithm is obtained in two steps. First, assuming that the Hermite coefficients are already known, the authors give an algorithm for recovering the parameters. This algorithm is non-robust since in reality the coefficients are not known, and we have to estimate them (with some errors) from the training set. In the second step the authors give the robust algorithm, that uses an unbiased estimator for the Hermite coefficients obtained from the training set. The algorithm relies on Jennrich’s algorithm for tensor decomposition.


**Limitations And Societal Impact:**

Yes

**Main Review:**

The problem studied in the paper is natural and interesting. Indeed, a similar problem has been already studied in several prior works, but the prior results do not handle the standard setting of ReLU activations with bias terms. The assumptions in the paper are reasonable. Some assumptions on the input distribution and on the weight matrix are required in order to obtain an efficient algorithm, and all prior results also include similar assumptions. Moreover, the fact that the assumption on W^{\odot \ell} holds w.h.p. in the smoothed-analysis setting suggests that it is likely to hold in real-world settings.

The paper considers the realizable setting, where we have y_i=f(x_i) for a target network f. However, the paper does not discuss whether the results can be extended to a non-realizable setting, where we have y_i=f(x_i)+\xi_i for some noise \xi_i. The robust algorithm tolerates errors in the estimation of the Hermite coefficients, which occur since we estimate the coefficients using a polynomial number of samples. What happens in the non-realizable setting where the errors in the estimates of the coefficients are also affected by \xi_i ?

Overall, I think that the results in the paper are significant and recommend acceptance.

Some minor comments:
- Line 56: “show give”
- Line 104: v_u -> v_i
- In lines 147-148 the authors mention recent SQ lower bounds for learning deep networks with Gaussian marginals. There is also a recent hardness result for improper learning of neural networks with Gaussian marginals (Amit Daniely and Gal Vardi, From Local Pseudorandom Generators to Hardness of Learning), which holds for all algorithms.
- Line 468: “if and if”
- Lines 490 and 495: “A.5” appears as a subscript
- Line 829: \tilde{W} instead of \hat{W}


**Time Spent Reviewing:**

6

---

> ### Author Response · Authors · 2021-08-10
> **Author response to reviewer pfsw**
>
> We thank the reviewer for the feedback and the reference to the work of Daniely and Vardi -- we will include this reference in the updated version of our paper.
>
> "What happens when there are errors?"
>
> We note that our algorithmic techniques are all already robust to a (small) inverse polynomial amount of noise (each of the individual components has such robustness). Under stronger assumptions about the errors e.g., i.i.d. random noise (e.g., zero mean Gaussian), we suspect that we can achieve much better guarantees that can be calibrated in terms of the signal-to-noise ratio. For the general non-realizable setting, the learning problem falls into the agnostic learning framework in which case there is strong evidence that even learning a single ReLU unit is computationally hard (see [GGJ+20, DKKZ’20]).

---

> > ### Comment · Reviewer_pfsw · 2021-08-31
> > **Response to the authors**
> >
> > Thank you for the response. After reading the other reviews and the authors' responses, I keep my (high) score.

---

### Official Review · Reviewer_MHqc · 2021-07-13

**Rating:** 7
**Confidence:** 4

**Summary:**

Essentially all existing provable guarantees for learning neural networks over Gaussian inputs focus on activations without a bias term. This submission gives a polynomial-time algorithm for (parameter and PAC) learning one hidden layer neural networks with ReLU activations and nonzero biases over Gaussians when the hidden weight matrix is well-conditioned. In fact their guarantee is more general: even if the hidden weight matrix is ill-conditioned/not full rank, if for some degree k, the k-th tensor powers of the hidden weight vectors are well-conditioned, then their algorithm runs in time exponential in \ell.

Like many previous algorithms in this literature, theirs is based on tensor decomposition of suitable Hermite coefficients of the network. Just as in the zero-bias case, the k-th Hermite coefficient of the network turns out to be the low-rank tensor given by a linear combination of k-th tensor powers of the hidden weight vectors. But unlike in the zero-bias case where the coefficients are given by the output layer weights, in the nonzero-bias case, each of these coefficients is additionally multiplied by a factor given by a certain function g_k(b_i) of the corresponding bias b_i (see Lemma 3.4). These coefficients can be quite small, and this poses several technical challenges that this paper overcomes.

As a corollary of their result and known results on the condition number of collections of tensor powers of *smoothed* vectors, they also get a guarantee for learning one hidden layer networks with biases in a smoothed setting.


**Limitations And Societal Impact:**

The authors adequately address the limitations of their work. The result is primarily of theoretical interest, and I don't see any potential negative societal impact.

**Main Review:**

The general approach of tensor decomposing Hermite coefficients is by now very standard in this line of  work. The main technical innovation in this paper is showing how to deal with components in the tensor which have small coefficient because g_k(b_i) is small.

The idea is fairly simple. g_k(b_i) turns out to be, up to a sign, He_{k-2}(b_i) * gamma(b_i), where gamma is the pdf of N(0,1), so g_k(b_i) can be small either because He_{k-2}(b_i) is small, or because |b_i| is very large so that the corresponding neuron barely ever or almost always activates. The latter case is not too hard to deal with, because the sum of all neurons for which |b_i| is large essentially behaves like a linear function. Handling the former case is arguably the most interesting part of the paper.

Their strategy is to aggregate estimates for the hidden weight vectors by decomposing *two tensors*, namely the Hermite coefficients at degree k and k+2; the punchline is that at least one of He_{k-2}(b_i) and He_k(b_i) must be non-negligible, so provided |b_i| is not too large, the coefficient of the corresponding rank-1 component in at least one of the two tensors will be sufficiently large.

There is a decent amount of technical work that goes into implementing these ideas, and overall it's a solid, well-written result. The particular problem they address was a well-known drawback of existing papers on learning shallow nets over Gaussians, so it's nice that a slight modification of existing tensor-based approaches suffices to handle nonzero biases in the well-conditioned case. Another nice feature is that they make explicit the observation that these tensor-based approaches don't need to assume the weight vectors are well-conditioned, but merely that their tensor powers are. This is a folklore fact to people familiar with tensor decomposition, but to my knowledge I don't think it was ever pointed out in any of the papers on learning neural nets using tensor decomposition.

The only criticism is that the practical significance is limited, given the Gaussianity assumption and the way in which it's used by looking at Hermite coefficients, but I think it's a solid theoretical contribution nevertheless.

Minor comments:
- Line 31: "designing algorithm" -> "designing algorithms"
- Line 32: the citation of [GK19] is a bit odd as that paper focuses on x's from the unit ball
- Line 56: "we first show give" -> "we first give"
- Line 98: "we wll" -> "we will"
- Line 147: "Given the recent statistical query" -> "Given the recent correlational statistical query"
- Line 151: again, not sure if [GK19] is relevant here
- Line 180: there should be a pointer to A.1 for the definition of this poly_ell notation
- Supplement eqs. (18), (24), (25): check formatting of brackets
- Supplement eq. (21) ||...|| should be |...|, and similarly for subsequent lines

**Time Spent Reviewing:**

5

---

> ### Author Response · Authors · 2021-08-10
> **Author response to reviewer MHqc**
>
> We thank the reviewer for their detailed comments, and feedback. We will update the final version accordingly.
>
> On the gaussianity assumption:
>
> We note that from a theoretical perspective even learning of ReLU networks under the Gaussian distribution is a challenging problem and most recent works on the topic make the same assumptions. Going beyond Gaussians is a natural direction for future work.
> One potential approach is to consider appropriate orthogonal polynomials as in [JSA15] instead of Hermite polynomials, and learn a network from those coefficients.

---

> > ### Comment · Reviewer_MHqc · 2021-08-29
> > **Thanks!**
> >
> > Thanks for the response! I totally agree that even fully understanding the Gaussian case is a challenging and important problem. In any case, I remain convinced that this paper would be a great addition to the program and will leave my score unchanged.

---

### Official Review · Reviewer_4QxN · 2021-07-16

**Rating:** 6
**Confidence:** 2

**Summary:**

This paper improves upon existing work on learning two-layer relu networks by considering a bias term. The paper presents polynomial time efficient algorithms for this learning task, states the required assumptions, and provides theoretical guarantees.

**Limitations And Societal Impact:**

Not applicable.

**Main Review:**

1. Regarding the motivation of studying the learning problem in the presence of the bias term, it is stated in the introduction that “The diminished expressivity of neural networks without the bias terms leads to the following compelling question:..”. By concatenating a 1 to the feature vector x, we can achieve the same expressivity without a bias term (though this new vector won’t be Gaussian now). I think the motivation for the bias term needs further justification.

2. Is learning planted weights useful in practice? What would be some applications? In my opinion, a discussion on this would improve the paper for a wider audience.

3. I think this paper is well-written and the assumptions and results are clearly stated. My biggest concern is around the motivation of this work.

4. I think some numerical simulation results to verify the theoretical results could also be useful.


**Time Spent Reviewing:**

3

---

> ### Author Response · Authors · 2021-08-10
> **Author response to the Reviewer 4QxN**
>
> We thank the reviewer for your comments and clarification questions. We address two of the specific questions.
>
> “Why include bias term instead of concatenating 1 to x?”
>
> All known results on learning neural networks (of depth 2 and above) assume  distributional assumptions about the input x (e.g., they are Gaussian, i.i.d and symmetric etc.). While one can simulate the bias term by introducing an extra co-ordinate that is always 1, this alters the distribution of x. The distribution of x is no longer a standard Gaussian (it is not even symmetric), and all the known algorithmic results fail under this setting. In particular, the specific challenges of how the existing algorithms (that rely on the 4th order Hermite coefficients) fail with bias term -1 or 1 continues to remain true. Similarly the large bias terms continue to remain a problem. Hence a bias term (or the equivalent concatenated input) is more challenging than the problem with no bias terms.
>
> “Is learning planted weights useful in practice? What would be some applications?”
>
> We view our main contribution as a polynomial time algorithm that achieves the weaker goal of learning a depth-2 ReLU network with low (test) error, under mild conditions. As the reviewer notes, our algorithm also recovers parameters of the units whose the bias terms are not too large in magnitude (under appropriate assumptions). Recovering such planted weights could have applications for interpretability and explainability. On the technical front, we remark that for many high-dimensional learning problems (e.g., mixture models like GMMs) efficient algorithmic techniques also perform the stronger task of parameter recovery. We will include a discussion in the final version.
>
> “some numerical simulation results to verify the theoretical results could also be useful”
>
> We thank the reviewer for this suggestion. While the main contribution of this paper is theoretical in flavor, we expect that insights from our work (e.g., Proposition B.2 in the Appendix) to be practically useful. Developing more practical variants of the proposed algorithm, and their empirical evaluation are natural topics for followup work.

---

> > ### Comment · Reviewer_4QxN · 2021-08-26
> > **Response to the authors**
> >
> > I have read the authors' response and thank the authors for their response. I keep my evaluation the same.

---

### Decision · Program_Chairs · 2021-09-27

**Decision:**

Accept (Poster)

**Comment:**

A solid progress in a well studied line of research